# POLICY-BASED SENTENCE SIMPLIFICATION: REPLACING PARALLEL CORPORA WITH LLM-AS-A-JUDGE

## ABSTRACT

Sentence simplification aims to modify a sentence to make it easier to read and understand while preserving the meaning. Different applications require distinct simplification policies, such as replacing only complex words at the lexical level or rewriting the entire sentence while trading off details for simplicity. However, achieving such policy-driven control remains an open challenge. In this work, we introduce a simple yet powerful approach that leverages Large Language Model-as-a-Judge (LLM-as-a-Judge) to automatically construct policy-aligned training data, completely removing the need for costly human annotation or parallel corpora. Our method enables building simplification systems that adapt to diverse simplification policies. Remarkably, even small-scale open-source LLMs such as Phi-3-mini-3.8B surpass GPT-4o on lexical-oriented simplification, while achieving comparable performance on overall rewriting, as verified by both automatic metrics and human evaluations. The consistent improvements across model families and sizes demonstrate the robustness of our approach[1].

## 1 INTRODUCTION

Sentence simplification could benefit users with reading difficulties, such as second-language (L2) learners and people with reading impairments (e.g., dyslexic individuals), by making text easier to read and understand (Alva-Manchego et al., 2020b). It involves a series of edits, such as lexical paraphrasing, sentence splitting, and removing irrelevant details (Xu et al., 2015). The preferred edit policy, i.e., permissible or appropriate edits in given texts, varies significantly depending on the target audience. In L2 education, one of the major application areas for simplification, previous work in both NLP and language education research has shown that the desired type and degree of simplification edits change depending on learner proficiency and readability levels (Agrawal et al., 2021; Zhong et al., 2020). Specifically, low- to intermediate-level learners benefit from a combination of lexical paraphrasing, structural modifications, and selective deletions to reduce cognitive load. In contrast, advanced learners benefit from lexical paraphrasing, which supports vocabulary acquisition (Chen, 2019), but they gain comparatively less from added cohesion or deletion (Hosoda, 2016; Zhong et al., 2020). Motivated by these findings, we introduce two distinct edit policies. As illustrated in Table 1, **overall-rewriting** simplification often combines lexical paraphrasing, structural modifications, and deletions to improve readability for intermediate-level language learners. In contrast, ~~advanced language learners may favor~~ **lexical-paraphrasing** ~~alone~~ (Paetzold & Specia, 2016; Li et al., 2025) adheres to the original sentence closely while supporting more efficient vocabulary acquisition for advanced learners.

Recent studies show that large proprietary LLMs such as OpenAI's ChatGPT models (OpenAI, 2023) achieve superior performance on simplification and often generate a mixture of diverse edit types (Kew et al., 2023; Heineman et al., 2023). However, their use in real-world applications such as language education is constrained by limited transparency and controllability. Running large open-source LLMs locally could be an alternative, but the heavy resource demands may make this impractical. Small-scale open-source LLMs present a more feasible option, yet adapting them with policy-driven simplification remains challenging. Key obstacles include: (1) the intrinsic limitations of LLMs, particularly smaller models, which are strong in overall quality but insensitive in following specific edit policies (Barayan et al., 2025); and (2) the scarcity of policy-specific parallel

---

[1]We will release our code and data after the paper gets accepted.

Table 1: Simplifications by our model under two edit policies (Phi-3-mini-3.8B (Abdin et al., 2024a)). We highlight the main simplification edits in each part of the sentence using different colors. Red: Deletions   Green: Paraphrasing   Blue: Split

| Source | Shade sets the main plot of the novel in motion when he impetuously defies that law, and inadvertently initiates a chain of events that leads to the destruction of his colony's home, forcing their premature migration, and his separation from them. |
|---|---|
| Overall-Rewriting | Shade defies the law and starts a chain of events that destroys his colony's home and forces them to leave early. He also separates from them. |
| Lexical-Paraphrasing | Shade starts the main story when he breaks the law on a whim, causing a series of events that destroy his colony's home and forces them to leave early, separating him from them. |

simplification corpora. Different from parallel texts for machine translation and summarisation that can be crawled from the web, sentences written in different readability levels are scarce. Manual construction of such a parallel corpus is prohibitively expensive. No existing studies provide an efficient way, in terms of both data and computational demands, for building simplification models adapted to predefined edit policies.

Reinforcement learning from human feedback (RLHF), introduced by OpenAI (Ouyang et al., 2022), has proven effective for aligning LLMs with human values. RLHF leverages human preference data rather than parallel corpora. However, collecting human preferences at scale is still costly. Alternatively, LLM-as-a-Judge can provide scalable and explainable feedback (Kocmi & Federmann, 2023; Song et al., 2024; Niu et al., 2024). Building on this, reinforcement learning from AI feedback (RLAIF) (Bai et al., 2022) appears promising to replace human preference with preferences generated by off-the-shelf LLMs (Tunstall et al., 2023; Cao et al., 2024; Lee et al., 2024).

In this work, we introduce a framework for policy-aligned sentence simplification that requires neither parallel corpora nor human supervision, while remaining computationally efficient with smaller LLMs. We focus on two distinct edit policies: lexical-paraphrasing and overall-rewriting. Our approach uses reasoning-capable LLMs as judges to automatically generate high-quality preference data under each policy. These data are then used to fine-tune open-source models, including Phi-3-mini-3.8B (Abdin et al., 2024a), Qwen2.5-7B (Yang et al., 2025b), Llama3.1-8B (Grattafiori et al., 2024), and Qwen2.5-14B (Yang et al., 2025b), via light-weight preference optimization algorithms (Xu et al., 2024; 2025). Our method significantly enhances the policy alignment capabilities of small-scale LLMs, enabling them to surpass GPT-4o on lexical-paraphrasing, and achieve comparable performance on overall-rewriting, as verified by both automatic metrics and human evaluation.

## 2 Edit policy alignment with LLM-as-a-Judge

### 2.1 Problem Definition

We train a simplification model using a decoder-only language model $\pi_\theta$ parameterized by $\theta$. Let $\mathcal{P}$ denote a set of simplification policies (e.g., *overall-rewriting*, *lexical-paraphrasing*), and let $p \in \mathcal{P}$ be a specific policy. Let $\mathcal{X}$ be a finite set of source sentences. For $x \in \mathcal{X}$, let $y^\star(x, p)$ be the (latent) ideal simplification under policy $p$. Our goal is to learn $\pi_\theta$ such that

$$\max_\theta \ \mathbb{E}_{x \sim \mathcal{X}} \big[ \log \pi_\theta \big( y^\star(x, p) \mid x \big) \big]. \tag{1}$$

However, $y^\star(x, p)$ is unobserved. Prior work approximated it relying on human-written simplifications and optimized $\pi_\theta$ through supervised fine-tuning (Scarton & Specia, 2018; Martin et al., 2020). In contrast, we build a preference dataset by LLMs, and optimize $\pi_\theta$ using preference optimization.

### 2.2 Method

Figure 1 illustrates our three-step framework for each policy.

**Step 1: Candidate Pool for Preference Data** We begin with a collection of $N$ source sentences $\mathcal{X} = \{x_1, x_2, \ldots, x_N\}$, and use LLMs to generate candidate simplifications. Diversity is crucial

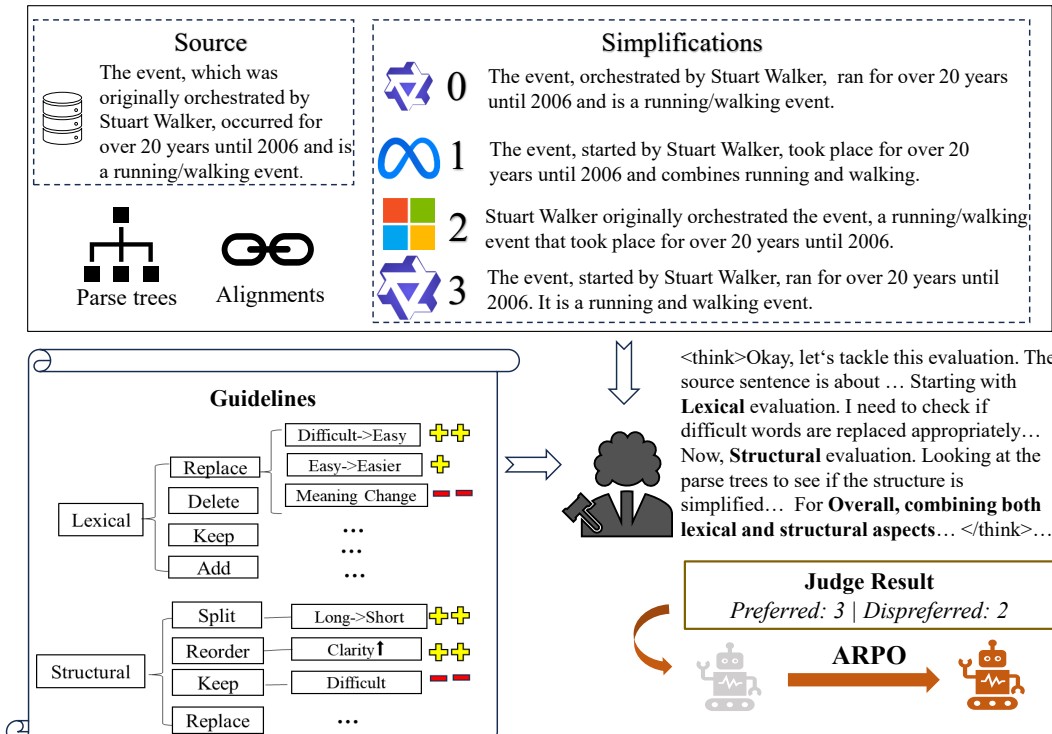

Figure 1: Overview of our framework. We collect simplifications from four LLMs: Qwen2.5-7B (Yang et al., 2025b), Llama3.1-8B (Grattafiori et al., 2024), Phi4-14B (Abdin et al., 2024b), and Qwen3-32B (Yang et al., 2025a). Based on the guidelines (++/--: high reward/penalty, +/-: moderate reward/penalty), the reasoning judge LLM evaluates along three dimensions: lexical, structural, and overall. Depending on the edit policy, we use either lexical (for lexical-paraphrasing) or overall (for overall-rewriting) preference to train LLMs.

to effectively distinguish preferred outputs that align with the target policy from those that deviate from it. Previous studies have shown that models of different families and sizes exhibit distinct performance and editing behavior (Heineman et al., 2023; Kew et al., 2023; Wu & Arase, 2025). Motivated by this, we construct the candidate pool using a set of $K$ LLMs, $\mathcal{M} = M_1, \ldots, M_K$, varying from different families and sizes. For each source sentence $x_i \in \mathcal{X}$ and policy $p$, every model $M_k \in \mathcal{M}$ generates one candidate simplification $y_{i,k}$, yielding a total of $K$ candidates per source sentence. The simplification pool for each $x_i$ is defined as:

$$\mathcal{C}(x_i) = \{y_{i,k} \mid y_{i,k} \sim M_k(x_i, s_p), \ k = 1, \ldots, K\}, \quad i = 1, \ldots, N \tag{2}$$

where $s_p$ is the natural-language instruction that describes policy $p$.

**Step 2: LLM-as-a-Judge** Lexical-paraphrasing encourages (a) minimal edits to preserve sentence structure and (b) replacing complex words with simpler alternatives. Overall-rewriting encourages broader edits at both lexical and structural levels to enhance simplicity. These nuanced heuristics can be included in prompts, allowing LLMs to follow them with ease. We employ a reasoning LLM as the judge, selected for its strong performance on complex reasoning tasks. The model is guided by carefully designed principles that specify edit types, their effects, and associated rewards or penalties (see Appendix A.2.1 for prompts).

- **Lexical Principles:** We define four edit operations—*replace*, *delete*, *keep*, and *add*. For example, simplifying a complex word receives a high reward, while replacing an already simple word yields only a moderate reward.

- **Structural Principles:** Similarly, we define four edit operations—*split*, *reorder*, *keep*, and *replace*. Structural transformations are rewarded if they improve readability or conciseness without changing meaning. Unnecessary or unhelpful modifications are penalized.

To support these judgments, we provide the lexical judge with word alignment between the source and simplifications derived using OTAlign (Arase et al., 2023) and the structural judge with syntactic parse trees for each sentence extracted using Qwen3-32B (Yang et al., 2025b), respectively.

For each source sentence $x_i \in \mathcal{X}$, the judge LLM selects a preferred candidate $y_w^{(i)}$ and a dispreferred candidate $y_l^{(i)}$ from the candidate pool $\mathcal{C}(x_i)$ according to our guidelines $\mathcal{G}$:

$$(y_w^{(i)}, y_l^{(i)}) = J(x_i, \mathcal{C}(x_i, p), \mathcal{G}), \quad i = 1, \ldots, N. \tag{3}$$

This procedure yields a preference dataset $\mathcal{D} = (x^{(i)}, y_w^{(i)}, y_l^{(i)})_{i=1}^{N}$ for each policy $p$.

**Step 3: Preference Optimization** Preference optimization has emerged as a powerful post-training paradigm for aligning LLMs with human preferences, typically formatted as {*input, preferred output, dispreferred output*}. It was first popularized by InstructGPT (Ouyang et al., 2022) using proximal policy optimization (PPO) (Schulman et al., 2017). However, PPO suffers from instability, high variance, and complexity, as it requires a reward model and online reinforcement learning. To address these limitations, Direct Preference Optimization (DPO) (Rafailov et al., 2023) was proposed as a lightweight alternative, removing the need for training an explicit reward model.

Building on DPO, Contrastive Preference Optimization (CPO) (Xu et al., 2024) further improves memory efficiency through a simpler preference loss combined with a behavior cloning loss, reducing reliance on a reference model required by DPO. CPO has shown strong performance on short-text generation tasks such as machine translation. In parallel, Simple Preference Optimization (SimPO) (Meng et al., 2024) also offers a reference-free but more stable formulation, incorporating length normalization and a target reward margin. CPO and SimPO can be combined to CPO-SimPO for improved performance and stability[2].

In this work, we adopt Adaptive Rejection Preference Optimization (ARPO) (Xu et al., 2025) to optimize $\pi_\theta$ with our preference dataset $\mathcal{D}$. ARPO is a CPO variant designed to mitigate its tendency to overly penalize dispreferred responses that are only marginally worse than preferred ones. Following CPO-SimPO, we integrate the SimPO loss into ARPO.

## 3 RELATED WORK

### 3.1 SENTENCE SIMPLIFICATION

Conventional sentence simplification relied on supervised fine-tuning (SFT) of sequence-to-sequence models using parallel corpora constructed from human-written simplifications. The two main resources are Simple English Wikipedia (SEW)[3] and Newsela (Xu et al., 2015). SEW provides simplified versions of Wikipedia[4] articles with fewer words and simpler grammatical structure. Newsela provides news articles, each professionally rewritten into up to five versions with varying readability levels. Sentence-level corpora are typically created by automatically aligning sentences between standard and Simple English Wikipedia articles, or across the multiple reading levels in Newsela (Alva-Manchego et al., 2020b). To compensate the limited amount of parallel corpora, Martin et al. (2022) crawled a large-scale pseudo-parallel sentences from the web and showed their effectiveness in building sentence simplification models. Nonetheless, these corpora are based on overall rewriting without adherence to a specific edit policy. Martin et al. (2020) showed that simple surface-level attributes such as sentence length or lexical difficulty can be controlled through prepended control tokens to the input. However, adaptation to various policies has been out of the scope of these previous studies. With the advent of LLMs, prompt-based techniques have largely surpassed earlier sequence-to-sequence model-based methods in overall simplification quality (Kew et al., 2023; Wu & Arase, 2025). However, prompting offers limited sensitivity to edit policy adaptation, particularly with smaller LLMs (Barayan et al., 2025).

Among studies on sentence simplification, the control of difficulty levels has been explored, which aims to simplify sentences to be appropriate for the target audience of the specific proficiency levels (Scarton & Specia, 2018; Horiguchi et al., 2024; Li et al., 2025). In particular, Li et al. (2025)

---

[2] https://github.com/fe1ixxu/CPO_SIMPO

[3] https://simple.wikipedia.org/wiki/Main_Page

[4] https://www.wikipedia.org/

employed reinforcement learning on LLM to control output difficult levels without parallel corpora. Nonetheless, these studies focus on the control within the specific type of edit policies (i.e., sentence difficulty), which may not extend to other types of policies.

## 3.2 SIMPLIFICATION EVALUATION METRICS

We propose LLM-as-a-Judge to evaluate how well simplification outputs align with the desired policy, which shares the goal with the evaluation metrics for sentence simplification. Before the era of LLMs, evaluation typically relied on high-quality human-written references. Formally, given a source sentence $s$, a target simplification $t$, and one or more reference simplifications $r$, the task of evaluating simplification is to compute a score $q(s, t, r)$. Evaluation methods are considered reliable if they demonstrate high correlation with human ratings (Liu et al., 2025).

These metrics can be broadly categorized into *statistic-based* and *model-based*. The most widely adopted statistic-based metric is **SARI** (Xu et al., 2016), which assesses lexical edit (eg, add, delete) quality by comparing system outputs with both references and the source sentence. Unlike other statistic-based metrics such as BLEU (Papineni et al., 2002)—which tends to give high scores to simplifications that are close or even identical to the source—SARI has been shown to better capture edit quality and exhibit stronger correlation with human ratings (Xu et al., 2015; Sulem et al., 2018). A common model-based metric is **LENS** (Maddela et al., 2023), which is trained directly on human ratings of overall simplicity quality. LENS has shown strong correlation with overall simplicity quality and therefore rewards extensive edits (Huang & Kochmar, 2024; Wu & Arase, 2025).

The dependence on high-quality human references, which are expensive to collect, limits the applicability of these metrics. To address this, reference-free metrics have been developed. Among them, **LENS-SALSA** (Heineman et al., 2023) achieves high correlation with human judgments. It was trained on extensive fine-grained human annotations of edit types (e.g., substitutions, splits, deletions) and their effects (e.g., efficacy, severity), enabling it to approximate LENS scores in a reference-free manner. More recently, one research has begun exploring using LLM-as-a-Judge to assess overall simplification quality (Liu et al., 2025), aggregating the judgments of multiple LLMs to improve reliability. However, these metrics remain limited to assessing overall quality, as they lack mechanisms to adapt judgments to diverse simplification policies.

## 4 EXPERIMENTS

We evaluate our framework by comparing it with various baselines (Section 4.3). Both automatic (Section 4.4) and human (Section 4.5) evaluations confirm the effectiveness of our approach. Furthermore, the results on out-of-domain datasets (Section 4.6) validate our method's transferability.

### 4.1 IMPLEMENTATION

We constructed the dataset for preference optimization using the (only) source sentences of the CoEdit corpus (Raheja et al., 2023), which aggregates existing simplification parallel corpora.[5] For each source sentence, we collected outputs from four instruction-tuned LLMs, as illustrated in Figure 1, forming a quartet of simplifications: {0: Qwen2.5-7B, 1: Llama3.1-8B, 2: Phi4-14B, 3: Qwen3-32B}. As the LLM-as-a-Judge model, we employed Qwen3-32B, leveraging its flexible think/no-think mode. To ensure meaningful simplification, we apply heuristic filtering, such as removing very short source sentences that leave little room for edits. After filtering, we obtain a preference dataset of $8k$ triplets in the form {source, preferred simplification, dispreferred simplification} for each policy. We split it into $7k$ training and $1k$ development samples.

We apply preference optimization to four open-source instruction-tuned LLMs from different families and scales: Phi-3-mini-3.8B (denoted as 'Phi3-3.8B') (Abdin et al., 2024a), Qwen2.5-7B (Yang et al., 2025b), Llama3.1-8B (Grattafiori et al., 2024), and Qwen2.5-14B (Yang et al., 2025b).

---

[5]Note that we do not use the target sentences provided in CoEdit.

## 4.2 EVALUATION DATASETS AND METRICS

We evaluated all the methods on standard benchmark datasets using the associated metrics.

**Lexical-Paraphrasing policy:** We used the Turk test set (Xu et al., 2016), containing 359 source sentences paired with 8 human-written simplification references, constructed specifically for lexical-based simplification. We accordingly use **SARI** (Xu et al., 2015) to assess the quality of lexical edits.

**Overall-Rewriting policy:** We used the **ASSET** test set (Alva-Manchego et al., 2020a). While ASSET shares the same source sentences as Turk, it differs in its edit policy: more diverse edits, i.e., paraphrasing, deletion, and sentence splitting. Each source sentence pairs with 10 human-written references. To capture this broader range of edits, we evaluated with **LENS** (Maddela et al., 2023).

## 4.3 COMPARISON METHODS

We compare the proposed method (denoted as **PO_Think**) against four kinds of baselines.

**Base models (Vanilla)**: Instruction-tuned LLMs used directly, serving as a prompting-based baseline that reflects their innate policy-aligned ability without additional training.

**GPT-4o**: A state-of-the-art proprietary LLM (Wu & Arase, 2025), representing a strong prompting-based upper bound [6].

**SFT on human-written parallel corpora (Parallel)**: Models fine-tuned on policy-aligned parallel corpora written by human, representing the scenario where such data is available. We used dev sets of Turk and ASSET (size: $2k$)[7].

**LENS-SALSA Preference Optimization (LENS_SALSA)**: The dataset for preference optimization was created by the LENS-SALSA metric; candidates with the highest scores were regarded as preferred, while the lowest-scored ones as dispreferred. As it highly correlates with the LENS metric, this approximates the scenario optimizing LENS using preference optimization. Remind that LENS-SALSA is reference-free, however, its training requires extensive human annotations.

In addition, we evaluate two variants of our method as ablation studies:

**No-reasoning LLM-as-a-Judge (PO_No-think)**: To assess whether the reasoning process is crucial, we disabled the think mode when using Qwen3-32B as the judge for collecting preference data, keeping all other settings identical.

**SFT on Preferred Data (SFT_Think)**: As an alternative to preference optimization, we fine-tuned the model using only on the preferred candidates by LLM-as-a-Judge (with reasoning mode).

We used LoRA (Hu et al., 2022) for both PO and SFT training, with $\alpha = 32$ and $r = 16$. For ARPO, we implemented based on the authors' implementation[8] and paper (Xu et al., 2025). We set the total batch size to $128$ and the learning rate to $1e-4$. We set $\beta = 0.1$ and $\gamma = 1.5$ for the SimPO loss (Meng et al., 2024), while $\alpha$ is fixed to 1. For SFT, we used the LLaMA-Factory package (Zheng et al., 2024) with the learning rate as $2e-4$. All models were trained for one epoch on a single NVIDIA A6000 Ada 48GB GPU. We used the vLLM package (Kwon et al., 2023)[9] for inference with open-source LLMs. Inference for Qwen3-32B was conducted on a NVIDIA H100 SXM5 94GB GPU, while all other open-source LLMs were run on a NVIDIA A6000 Ada 48GB GPU. For non-reasoning models, we set the decoding parameters to temperature $= 0$, top-$p = 1.0$, and top-$k = -1$. For Qwen3-32B in think mode (only used for LLM-as-a-Judge), we followed the official settings[10], with temperature $= 0.6$, top-$p = 0.95$, and top-$k = 20$. For OTAlign, we used the supervised setting of the authors' implementation[11], using $\tau = 0.88$ and a threshold of $0.40$.

---

[6]The recent model GPT-5 shows performance very similar to GPT-4o, with scores of 41.2 SARI on Turk, 68.1 LENS, and 46.6 SARI on ASSET.

[7]Although both datasets provide multiple references per sentence, we used only one reference per sentence, as preliminary experiments showed performance degradation with multiple references during training.

[8]https://github.com/fe1ixxu/ALMA/tree/master

[9]https://github.com/vllm-project/vllm

[10]https://huggingface.co/Qwen/Qwen3-32B

[11]https://github.com/yukiar/OTAlign

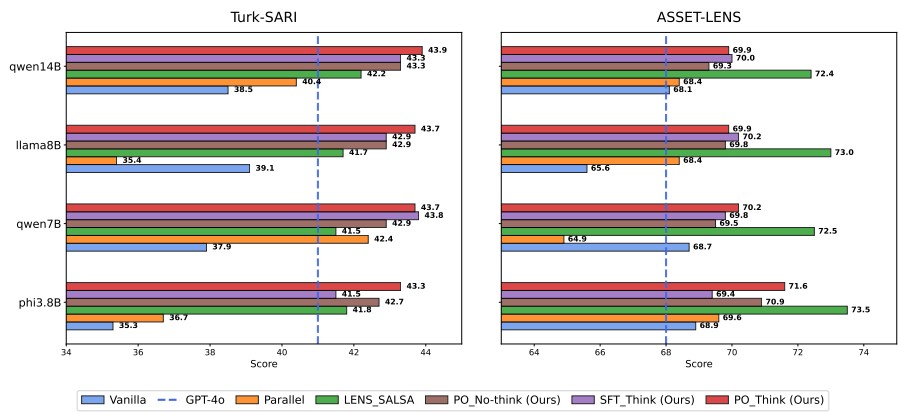

Figure 2: Automatic evaluation results. The higher the better.

Prompts used for collecting simplifications are described in Appendix A.2.2, and prompts for LLM-as-a-Judge are provided in Appendix A.2.1. $8k$ simplification candidates collection took about 3 minutes per model. LLM-as-a-Judge stage required about 7 hours to output judgments.

For inference during performance comparison, we used zero-shot settings for all models (as described in Appendix A.2.2), as we found that prompt-engineering strategies such as few-shot and iterative self-correction based on feedback did not consistently improve, and sometimes even degraded performance. We discuss the impact of these strategies in Appendix A.4. SARI was computed with the EASSE package (Alva-Manchego et al., 2019). LENS and LENS-SALSA were computed with the authors' implementation[12].

### 4.4 AUTOMATIC EVALUATION RESULTS

Automatic evaluation results are provided in Figure 2. SARI scores on ASSET dataset are provided in Figure 4 in Appendix.

**The proposed method outperforms GPT-4o with much smaller scale models.** For both simplification policies, our approach not only surpasses the vanilla models but also achieves results exceeding GPT-4o, as measured by both SARI and LENS metrics. On lexical-paraphrasing, SARI improves by +8.0 (Phi3-3.8B), +5.8 (Qwen2.5-7B), +4.6 (Llama3.1-8B), and +5.4 (Qwen2.5-14B). Even under the overall-rewriting policy, where LLMs already demonstrate strong performance due to their capacity for diverse edits, our method remains robust:+2.7 (Phi3-3.8B), +1.5 (Qwen2.5-7B), +4.3 (Llama3.1-8B), and +1.8 (Qwen2.5-14B). These results show that our approach reliably steers outputs toward policy-aligned simplifications.

**LLM-as-a-Judge consistently outperforms human-written parallel corpus.** Our method outperforms SFT on the human-written corpus (Parallel). Two factors contribute: (1) *Scalability:* LLM-as-a-Judge enables creating large-scale preference data easily and efficiently, whereas SFT is constrained by the scale of human efforts. (2) *Quality Control:* Human references are not always perfect and may even be surpassed by advanced LLMs (Xu et al., 2024; Liu et al., 2024). We observed the same issue in the Turk and ASSERT dev sets, where references sometimes violate simplification guidelines by deleting essential content, retaining difficult words, or offering only trivial changes (see Table 2). We further analyzed the effects of data sizes as shown in Figure 3. Even when models were trained on the same amount of data ($2k$ samples), our method consistently outperforms the Parallel baseline across all models and policies. This confirms (2), reflecting the high quality of our preference data.

**LENS_SALSA struggled on lexical-paraphrasing policy.** As expected, LENS_SALSA showed the highest LENS scores on the overall-rewriting policy (ASSET), where LENS is the evaluation metric. In contrast, it struggled with the lexical-paraphrasing policy (Turk). It is non-trivial to adapt LENS-SALSA for other simplification policies because it requires large-scale, careful human

---

[12]https://github.com/Yao-Dou/LENS

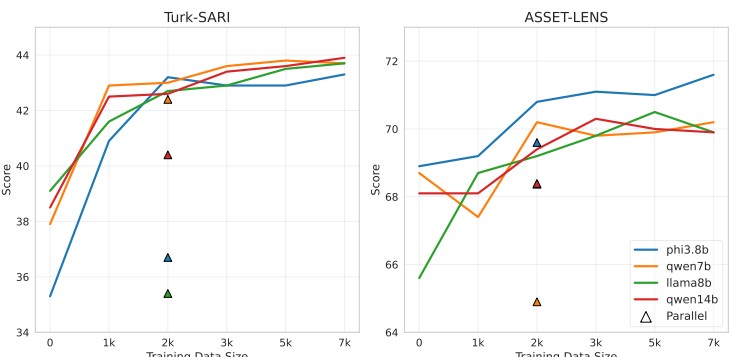

Figure 3: Impact of training sample size. Colored lines show models trained on our preference data; colored triangles show models trained on $2k$ human-written parallel data. Overlap in triangles is due to the nearly identical LENS scores from Qwen14B and Llama8B.

Table 2: Example of deficiencies in human references.

| Dataset | Content |
| --- | --- |
| Turk | **Source:** Approved indications for codeine include: Cough, though its efficacy in low dose over the counter formulations has been disputed. |
| | **Reference:** Approved reasons for codeine are a cough. |
| | **Issue:** Heavy information loss due to deletion. |

annotations. Different from LENS-SALSA, our method can easily adapt to a new simplification policy by adjusting LLM-as-a-Judge prompts. Furthermore, our human evaluation (Section 4.5) confirmed that the simplification qualities of LENS-SALSA and our method are competitive: not as significant as the LENS score indicates.

**Reasoning is crucial on LLM-as-a-Judge for simplification quality.** Training on reasoning-based preference data (PO_Think) consistently outperforms those from the non-reasoning mode (PO_No-think). This suggests that complex evaluations benefit from reasoning-enabled judges, yielding more reliable supervision and stronger policy alignment. We observed that reasoning and non-reasoning lead to divergent judgments. For both lexical-paraphrasing and overall-rewriting, the two modes disagree on more than $40\%$ of preference pairs. In about $20\%$ of cases, their judgments are directly opposite. That is, the candidate preferred by the reasoning mode is rejected by the non-reasoning mode, or vice versa. Table 3 presents the model-wise preference distributions. The reasoning process leads to generally more balanced distributions. For example, in lexical-paraphrasing, Qwen2.5-7B is preferred $48.4\%$ of the time in no-think mode, but only $31.0\%$ in think mode. We provide case studies in Appendix A.3 to verify whether LLM-as-a-Judge's judgments adhere to our guidelines.

**Preference optimization outperforms SFT.** On our preference optimization dataset, both SFT (SFT_Think) and preference optimization (PO_Think) achieve strong performance, yet the latter generally outperforms SFT. This finding may suggest that while preferred candidates are of high quality, incorporating pairwise preference signals rather than relying solely on positive examples leads to better policy alignment.

**The quality of simplification positively correlates with the scale of the preference optimization dataset.** We investigated how the size of the preference optimization dataset influences model performance. We sample subsets of $1k$, $2k$, $3k$, and $5k$ preference pairs from the training set, keeping all other settings fixed. Results are shown in Figure 3. The overall trend is clear: performance consistently improves as the training size increases. Models show a sharp gain once training size reaches $2k$, after which improvements become more gradual.

### 4.5 HUMAN EVALUATION

To assess the quality of sentence simplification, human evaluation is crucial. We conducted a human evaluation to assess whether the generated simplification adheres to the desired simplifica-

Table 3: Model-wise preference distribution (%) of Qwen3-32B under reasoning (Think) and non-reasoning (No-think) modes for lexical-paraphrasing and overall-rewriting policies. Each cell shows *Preferred / Dispreferred* ratios.

| Model | Lexical-Paraphrasing | | Overall-Rewriting | |
|---|---|---|---|---|
| | Think | No-think | Think | No-think |
| Qwen2.5-7B | 31.0 / 21.8 | 48.4 / 20.5 | 24.6 / 18.2 | 36.8 / 15.1 |
| Llama3.1-8B | 19.0 / 45.5 | 17.3 / 39.1 | 23.9 / 44.0 | 19.6 / 41.9 |
| Phi4-14B | 22.6 / 17.5 | 15.6 / 13.0 | 28.3 / 19.5 | 24.4 / 14.2 |
| Qwen3-32B | 27.4 / 15.2 | 18.7 / 27.4 | 23.2 / 18.3 | 19.2 / 28.8 |

Table 4: Human evaluation results using a 5-point Likert scale.

| Model | Mean |
|---|---|
| PO_Think (ours) | **4.13** |
| LENS_SALSA | 3.47 |
| GPT-4o | 3.65 |

(a) TURK

| Model | Mean |
|---|---|
| PO_Think (ours) | **4.12** |
| LENS_SALSA | 4.08 |
| GPT-4o | 3.93 |

(b) ASSET

tion policy. We annotated simplifications generated by our method (PO_Think) against two strong baselines: LENS_SALSA and GPT-4o using a 5-point Likert scale. Outputs from PO_Think and LENS_SALSA were generated by Phi3-3.8B model, the smallest in scale, but showed strong performance. The annotation was performed by one of the authors, who is familiar with the guidelines of Turk[13] and ASSET[14]. Consistent with the guidelines, higher scores on the Likert scale indicate stronger alignment with the simplification policies. We define scores above 4 as high alignment, scores between 3 and 4 as moderate alignment, and scores below 3 as low alignment.

As annotation targets, we randomly sampled 60 source sentences, yielding 180 source-simplification sentence pairs per policy, for a total of 360 pairs (2 policies × 3 models). The 180 sentence pairs within each policy were randomized so that the annotator would not know which model produced a given output. For each pair, the annotator was asked to assign a score from 1 to 5. The entire annotation process took approximately six hours. Results are provided in Table 4.

**Our method achieves high edit policy alignment, while LENS_SALSA may overfit to the LENS metric.** Our method achieves the highest mean score (above 4) on both Turk and ASSET, demonstrating strong alignment with edit policies and outperforming both baselines. Unlike the results under the LENS scores, where LENS_SALSA outperforms our method, human evaluation shows only marginal differences. This suggests that preference optimization with LENS_SALSA may cause overfitting to LENS. We observe that models trained with LENS_SALSA sometimes over-prioritize simplicity at the expense of accuracy, leading to lower human scores (see Table 5 for an example).

### 4.6 GENERALIZATION TO OUT-OF-DOMAIN SENTENCES

To assess out-of-domain generalization, we evaluated our models on two datasets outside the Wikipedia domain of ASSET and Turk: **SimPA**, drawn from the public administration domain (Scarton et al., 2018), and **Newsela**, derived from news domain (Xu et al., 2015; Zhang & Lapata, 2017).

**SimPA** contains 1,100 original sentences paired with two types of references: (1) lexical simplifications, and (2) overall simplifications combined with lexical and syntactic edits. We evaluated our lexical-paraphrasing model on the first type and our overall-rewriting model on the second.

**Newsela** consists of news articles with multiple simplified versions written by professional editors. The corpus was aligned from document-level to sentence-level; we used the test split, which contains 1,077 complex-simple sentence pairs. Because Newsela simplifications involve diverse editing operations, we evaluated using our overall-rewriting policy models.

---

[13]https://github.com/cocoxu/simplification/blob/master/HIT_MTurk_crowdsourcing/simplification_HIT_free_response.html

[14] https://github.com/facebookresearch/asset/blob/main/crowdsourcing/AMT_AnnotationInstructions.pdf

Table 5: Examples of simplifications and corresponding human evaluation scores.

(a) Lexical-Paraphrasing: PO_Think achieves high scores for effective paraphrasing ('overseen' → 'managed'), while GPT-4o loses some details and LENS_SALSA distorts meaning ('overseen' → 'given').

| System | Simplification | Score |
|---|---|---|
| Source | Formal minor planet designations are number-name combinations overseen by the Minor Planet Center, a branch of the IAU. | – |
| PO_Think | Formal minor planet names are number-name combinations managed by the Minor Planet Center, a part of the IAU. | 4 |
| LENS_SALSA | Formal minor planet names are given by the Minor Planet Center, a part of the IAU. | 1 |
| GPT-4o | Minor planet names and numbers are managed by the Minor Planet Center, part of the IAU. | 2 |

(b) Overall-Rewriting: LENS_SALSA sometimes over-prioritizes simplicity.

| System | Simplification | Score |
|---|---|---|
| Source | The term dorsal refers to anatomical structures that are either situated toward or grow off that side of an animal. | – |
| PO_Think | Dorsal means anatomical structures are on or grow from the back side of an animal. | 5 |
| LENS_SALSA | Dorsal means anatomical structures are on the top side of an animal. | 3 |
| GPT-4o | Dorsal refers to anatomical structures located on or growing from an animal's back side. | 5 |

Table 6: Out-of-domain evaluation results on SimPA and Newsela. We report results for two policy types: Lexical-Paraphrasing (L) and Overall-Rewriting (O). (GPT-4o scores: 28.6 SARI and 59.7 LENS on SimPA, and 60.9 LENS on Newsela.)

| Dataset-Policy-Metrics | System | Phi3.8B | Qwen7B | Llama8B | Qwen14B |
|---|---|---|---|---|---|
| SimPA-L-SARI | Vanilla | 22.3 | 26.5 | 27.7 | 27.5 |
| | PO_Think (ours) | **37.3** | **36.4** | **36.6** | **36.8** |
| SimPA-O-LENS | Vanilla | 60.1 | 56.0 | 59.8 | 58.7 |
| | PO_Think (ours) | **63.0** | **62.7** | **61.3** | **61.6** |
| Newsela-O-LENS | Vanilla | 62.6 | 59.1 | 61.0 | 62.3 |
| | PO_Think (ours) | **64.9** | **63.9** | **63.2** | **63.6** |

We applied our policy models directly using the same prompts and hyperparameters as in the wiki-domain experiments, without any retraining. Table 6 shows that, across both domains, our policy models outperforms the vanilla baselines, demonstrating that the learned policies can transfer effectively to different domains.

## 5 CONCLUSION AND FUTURE WORK

We propose a framework for adapting sentence simplification to various policies, which is critical for real-world applications. By leveraging LLM-as-a-Judge, our method removes the reliance on human-written parallel corpora and costly human annotations. Furthermore, our method consistently enhances the policy alignment of small-scale open-source LLMs, achieving comparable or even higher performance than the large proprietary LLM.

Our study focuses on sentence-level simplification, where large language models (LLMs) remain error-prone and struggle to consistently align with human preferences. Meanwhile, document-level simplification is also important for real-world applications but has been underexplored in prior work. To explore whether our framework can extend beyond the sentence level, we apply it to document-level simplification while keeping the overall three-step pipeline unchanged (see Appendix A.5). Preliminary results indicate that the framework generalizes well. Nonetheless, progress on document-level tasks is constrained by the lack of diverse, policy-aligned evaluation datasets and appropriate metrics. As a result, we were unable to further explore different edit policies at the document level. Developing such resources represents an important direction for future research.

Moreover, this study focuses on English simplification. Future studies could extend our framework to policy-driven simplification in other languages and explore its applicability beyond simplification, such as style transfer, lay-summarization, and other controllable text generation tasks.

## 6 ETHICS STATEMENT

This work adheres to the ICLR Code of Ethics. We do not identify any specific risks of ethical concern in this work. We used a Large Language Model (specifically, ChatGPT) to polish the writing of this paper. All content was independently drafted by the authors, and the model was used only for grammar correction and language refinement.

## 7 REPRODUCIBILITY STATEMENT

We ensure reproducibility of our results. All datasets and packages used are open-source and clearly referenced. Detailed settings and prompts are provided in Section 4.3, A.1, and A.2.

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

## A  APPENDIX

### A.1  INFERENCE SETTING

We used the vLLM package (Kwon et al., 2023)[15] for inference with open-source LLMs and OpenAI's API for GPT-4o. For non-reasoning models, we set the decoding parameters to temperature $= 0$, top-$p = 1.0$, and top-$k = -1$. For the reasoning model Qwen3-32B, we followed the official settings[16], with temperature $= 0.6$, top-$p = 0.95$, and top-$k = 20$. Inference for Qwen3-32B was conducted on a NVIDIA H100 SXM5 94GB GPU, while all other open-source LLMs were run on a NVIDIA A6000 Ada 48GB GPU.

---

[15] `https://github.com/vllm-project/vllm`
[16] `https://huggingface.co/Qwen/Qwen3-32B`

For OTAlign, we used the supervised setting of the authors' implementation[17], using $\tau = 0.88$ and a threshold of $0.40$. For evaluation, SARI was computed with the EASSE package (Alva-Manchego et al., 2019). LENS and LENS-SALSA were computed with the authors' implementation[18].

## A.2 PROMPTS

### A.2.1 LLM-AS-A-JUDGE

The prompt used for LLM-as-judge consists of a detailed set of evaluation guidelines and three in-context examples. Figure 7 shows the full guideline, which includes the explanation of the provided materials, task description, evaluation principles, and instructions for formatting the output. The 3-shot examples are illustrated in Figure 8, 9, and 10. In each example, the input contains a source sentence along with four simplification candidates, accompanied by their corresponding word alignments and syntactic parses. The output includes evaluation analysis and decisions across lexical, structural, and overall dimensions, documented by the authors.

### A.2.2 SIMPLIFICATION GENERATION

We use prompts for simplification generation at three stages: candidate collection, fine-tuning, and evaluation. The prompts are from a previous study (Wu & Arase, 2025) and illustrated in Figure 5. They are carefully designed and validated on the ASSET and Turk development sets. Each prompt contains an instruction followed by three in-context examples, which we employ for candidate collection. For fine-tuning (including both SFT and ARPO) and evaluation on target LLMs, we only use the system instruction part of the prompts.

### A.2.3 PARSE

We use a 1-shot prompt to obtain parses, as shown in Figure 6. The model receives a system prompt specifying the task and a single input–output demonstration to illustrate the expected format.

## A.3 CASE STUDIES

To better understand the strengths and limitations of reasoning-based judgment, we present representative examples comparing the outputs of the judge in *think* (reasoning) and *no-think* (non-reasoning) modes. Overall, we find that think mode adheres to the guidelines more faithfully, identifies, and rewards high-quality paraphrases. In contrast, no-think mode often favors conservative strategies such as retention or deletion, even when these conflict with policy objectives.

### A.3.1 EXAMPLES FROM LEXICAL-PARAPHRASING

**Source:** In seclusion, Janu became happy as she was away from the others that alienated her.

| ID | Candidate Output |
|----|------------------|
| 0 | In seclusion, Janu became happy as she was away from the people who alienated her. |
| 1 | Janu was happy in seclusion, away from her alienating friends. |
| 2 | In seclusion, Janu became happy because she was away from the people who alienated her. |
| 3 | In seclusion, Janu became happy because she was away from the people who made her feel left out. |

**Preferences:** Think → prefer 3, disprefer 1; No-think → prefer 0, disprefer 3.

In this example, the complex word *'alienated'* is retained in 0, 1, and 2, but paraphrased into the simpler phrase *'made her feel left out'* in 3. This is a high-quality paraphrasing. The reasoning judge correctly rewarded 3, explicitly noting in its reasoning chain:

> `<think>` ... *This is a paraphrase that simplifies the complex word "alienated" into a more straightforward phrase...* `</think>`

---

[17]https://github.com/yukiar/OTAlign
[18]https://github.com/Yao-Dou/LENS

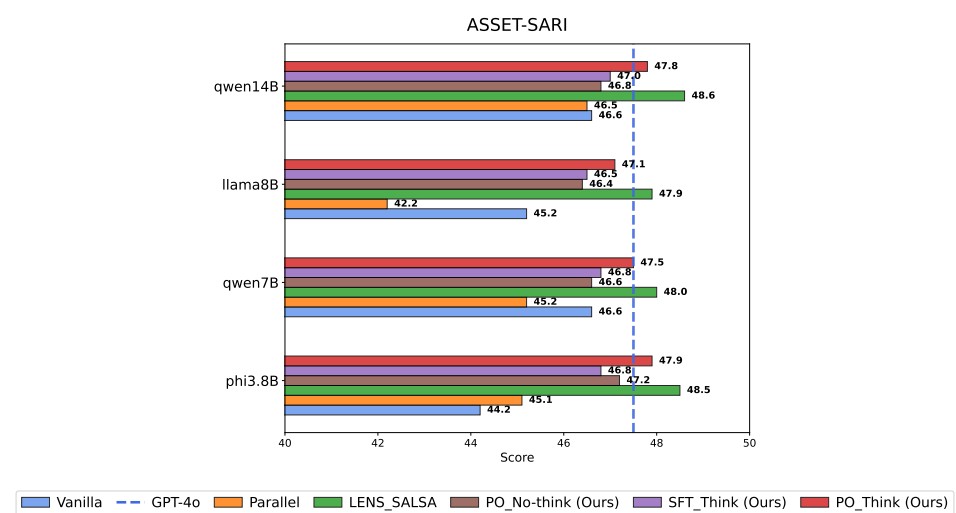

Figure 4: SARI scores on ASSET. The higher the better.

In contrast, the non-reasoning judge favored 0, failing to recognize effective paraphrasing.

### A.3.2 EXAMPLES FROM OVERALL-REWRITING

**Source:** It is a salt consisting of ammonium ions and dichromate ions.

| ID | Candidate Output |
|----|-----------------|
| 0  | It consists of ammonium and dichromate ions. |
| 1  | It's a salt made of ammonium and dichromate ions. |
| 2  | It is a salt made of ammonium ions and dichromate ions. |
| 3  | It is a salt made of ammonium and dichromate ions. |

**Preferences:** Think → prefer 2, disprefer 0; No-think → prefer 0, disprefer 3.

**Source:** They're safe, but given the depravity of these two, they'll roast alongside me.

| ID | Candidate Output |
|----|-----------------|
| 0  | They're safe, but these two are depraved, so they'll roast with me. |
| 1  | They're safe, but these two horrible people will be punished right along with me. |
| 2  | They're safe, but these two will roast alongside me. |
| 3  | They are safe, but because of how bad these two are, they will roast with me. |

**Preferences:** Think → choose 3, reject 2; No-think → choose 2, reject 1.

In the first example, the reasoning judge rewarded 2, which preserves the word *'salt'* while simplifying *'consisting of'* → *'made of'*. The non-reasoning judge favored 0, which deletes the easy word *'salt'* and loses important information. In the second example, the reasoning judge favored 3, which paraphrases *'given the depravity'* as *'because of how bad'*, a clearer and simpler expression that retains meaning. The non-reasoning judge chose 2, which deletes this information, discarding semantic nuance.

### A.3.3 LIMITATIONS OF REASONING JUDGES

Despite their benefits, reasoning judges are not flawless. We observed cases where the overall preference decision was correct, but word-level difficulty judgments were inaccurate.

**Source:** BRICS is the acronym coined for an association of five major emerging national economies: Brazil, Russia, India, China and South Africa.

| Label | Sentence |
|---|---|
| Prefer | BRICS is the name for a group of five major emerging economies: Brazil, Russia, India, China, and South Africa. |
| Disprefer | BRICS is the acronym for an association of five major emerging countries: Brazil, Russia, India, China and South Africa. |

Here, the reasoning judge reasonably preferred the simplification that replaced *'acronym coined for'* with *'name for'*. However, its reasoning chain incorrectly classified *'coined'* as an easy word:

> `<think>` *... maybe a moderate reward since 'coined' was an easy word replaced by simpler structure...* `</think>`

According to Common European Framework of Reference for Languages (CEFR)[19], *'coined'* is simple as a noun (payment object) but difficult as a verb (to invent). Future work could explore integrating external linguistic resources, such as CEFR-based wordlists or lexical databases, into reasoning judges to enhance their sensitivity to word difficulty in different contexts.

## A.4 PROMPT ENGINEERING VS. TRAINING

LLMs are known to be prompt-sensitive. To examine whether prompt engineering alone can achieve reliable policy-aligned simplification, we evaluated two commonly used strategies: **few-shot prompting** and **iterative feedback**. Few-shot prompting uses three in-context examples sampled from the ASSET and Turk development sets (see Figure 5). Iterative feedback includes automatic judgment and self-correction: using the same LLM-as-a-Judge and evaluation guidelines introduced in our framework, we generated feedback on model's output and requested the model to do self-correction based on the feedback for two rounds (*Iter 1* and *Iter 2*). See Figure 11, 12, 13 and 14 for prompts in this setting. Results are shown in Table 7.

**Our method consistently outperforms prompt-engineering strategies.** Across all model sizes and both policies, preference-trained models in the zero-shot setting achieve the strongest performance. The only marginal exception appears in overall rewriting on Llama3.1-8B, where our model performs on par with the few-shot baseline. Overall, the improvements delivered by our method are substantial and far more consistent than those obtained through prompt engineering.

**Few-shot prompting is not a robust solution.** Few-shot prompting does not reliably improve performance. It even degrades performance on overall rewriting for Qwen2.5-7B ($-0.3$) and Qwen2.5-14B ($-1.6$), suggesting that in-context examples may introduce biases. Moreover, the benefits of few-shot prompting vary widely across models: On lexical paraphrasing, it yields a gain of 6.9 SARI for Phi3-3.8B but only 1.6 for Llama3.1-8B. On overall rewriting, it improves Llama3.1-8B by 4.4 LENS but degrades both Qwen models. In contrast, after our preference training, all models reach similarly strong performance levels: above 43 SARI for lexical paraphrasing and around 70 LENS for overall rewriting. This indicates that our method provides more stability across architectures.

**Iterative feedback cannot reliably handle different policies.** Iterative feedback improves performance on lexical paraphrasing but consistently degrades performance on overall rewriting for every model tested. We found that iterative feedback makes models more conservative, resulting in fewer meaningful simplification edits. This suggests that feedback alone is insufficient to guide LLMs toward distinct and policy-aligned simplification behaviors.

## A.5 DOCUMENT-LEVEL SIMPLIFICATION

Compared with sentence-level simplification, document-level simplification introduces additional challenges such as maintaining global coherence, discourse structure, and coreference consistency (Vásquez-Rodríguez et al., 2023; Wilkens et al., 2020). To investigate whether our proposed

---

[19]`https://englishprofile.org/?menu=evp-online`

Table 7: Comparison between prompt engineering and our method. Simplification inference for iterative feedback and PO_think is in the zero-shot setting. Best scores are in bold, and second-best are underlined.

| Method | Phi3.8B | Qwen7B | Llama8B | Qwen14B |
|---|---|---|---|---|
| Vanilla-zs | 35.3 | 37.9 | 39.1 | 38.5 |
| Vanilla-fs | 42.2 | 41.0 | 40.7 | 42.6 |
| Iter 1 | 38.9 | 39.7 | 42.4 | 41.8 |
| Iter 2 | 39.8 | 40.6 | 42.8 | 42.3 |
| PO_think (Ours) | **43.3** | **43.7** | **43.7** | **43.9** |

(a) Lexical paraphrasing (Turk-SARI).

| Method | Phi3.8B | Qwen7B | Llama8B | Qwen14B |
|---|---|---|---|---|
| Vanilla-zs | 68.9 | 68.7 | 65.6 | 68.1 |
| Vanilla-fs | 70.0 | 68.4 | **70.0** | 66.5 |
| Iter 1 | 66.6 | 63.7 | 64.3 | 64.8 |
| Iter 2 | 66.0 | 63.6 | 64.0 | 65.7 |
| PO_think (Ours) | **71.6** | **70.2** | 69.9 | **69.9** |

(b) Overall rewriting (ASSET-LENS).

framework can be extended beyond the sentence level, we adapt it to document level while keeping the overall three-step pipeline unchanged. We only modified the LLM-as-a-Judge guidelines to account for document-level requirements. The following summarizes our experiment setup.

**Step 1: Candidate Pool for Preference Data** We used around $1.8k$ complex documents from the Newsela dataset (Xu et al., 2015) (same dataset described in Section 4.6, but we used samples at the document level) as inputs and generated multiple simplifications from the same set of LLMs. Importantly, we did not include Newsela reference simplifications in the candidate pool.

**Step 2: LLM-as-a-Judge** We expanded the evaluation guidelines to include aspects critical for document-level simplification: simplicity, discourse relations, coreference resolution, and global coherence. Similarly to the design for sentence-level, the evaluation principles specify edit types, their effects, and associated rewards or penalties (refer to Figure 16 for the complete guidelines). The same LLM-as-a-Judge model then selected preferred and dispreferred candidates.

**Step 3 Preference Optimization:** We applied the same preference-optimization procedure to train preference-optimized models using the document-level preference dataset. Experiments were conducted on Qwen2.5-7B and Llama3.1-8B [20].

We evaluated the models on 200 held-out Newsela documents with their corresponding reference simplifications [21]. Two automatic metrics designed for document-level simplification were used:

- **Agg-LENS** (Maddela & Alva-Manchego, 2025), the SOTA document-level simplification metric with the strongest correlation to human judgments.
- **Agg-BERTScore** (Maddela & Alva-Manchego, 2025), which shows high consistency for assessing coherence in simplification.

We compare our models with zero-shot and few-shot prompting baseline. The instruction used is provided in Figure 15. The few-shot examples are taken directly from Newsela, using documents that do not appear in either the training or test sets [22]. As shown in Table 8, our method outperforms both zero-shot and few-shot, demonstrating that our framework generalizes effectively to document-level simplification.

---

[20]Phi3-3.8B supports only a 4k context window, which limits its applicability to document-level simplification. Training Qwen2.5-14B on long documents was not feasible given our available computational resource.

[21]We used the simplifications with the lowest grade level as the reference.

[22]We cannot include the examples in the paper due to Newsela's licensing restrictions. Please obtain access from https://solutions.newsela.com/.

Table 8: Document-level simplification results.

| Method–Prompt | Agg-LENS | Agg-BERT | Method–Prompt | Agg-LENS | Agg-BERT |
|---|---|---|---|---|---|
| Vanilla-zs | 37.1 | 15.1 | Vanilla-zs | 36.0 | 16.0 |
| Vanilla-fs | 39.5 | 17.0 | Vanilla-fs | 43.5 | 17.4 |
| po_think (Ours) | **46.2** | **25.0** | po_think (Ours) | **48.0** | **27.7** |

|  (a) Qwen2.5-7B  |  (b) Llama3.1-8B  |
|---|---|

```
System:
You are required to simplify the original sentence by using simpler concepts, words, or
phrases. Pleasekeep the meaning the same. Only provide one result.

User:
San Francisco Bay is located in the U.S. state of California, surrounded by a
contiguous region known as the San Francisco Bay Area, dominated by the large cities
San Francisco, Oakland and San Jose.

Assistant:
San Francisco Bay is located in the U.S. state of California, surrounded by a
contiguous region known as the San Francisco Bay Area, influenced by the large cities,
San Francisco, Oakland and San Jose.

User:
The book chronicles events which take place in the fictional space colony of Windhaven.

Assistant:
The book chronicles events which take place in the space colony of Windhaven.

User:
Some academic journals do refer to Wikipedia articles, but are not elevating it to the
same level as traditional references.

Assistant:
Some academic journals do refer to Wikipedia articles, but are not using it to the same
level as common references.

User:
{Input}
```

(a) Lexical-Paraphrasing

```
System:
You are required to simplify the original sentence by applying different
transformations. Please keep the meaning the same. Only provide one result.

User:
Rollins retired in 1962 and opted to become a coach.

Assistant:
Rollins retired in 1962. He then chose to become a coach.

User:
Tourism is concentrated in the mountains, particularly around the towns of Davos /
Arosa, Laax and St. Moritz / Pontresina.

Assistant:
Tourism takes place in the mountains around the towns of Davos / Arosa, Laax and St.
Moritz / Pontresina.

User:
First Fleet is the name given to the 11 ships which sailed from Great Britain on 13 May
1787 with about 1,487 people to establish the first European colony in New South Wales.

Assistant:
11 ships sailed from Great Britain on 13 May 1787 carrying about 1,487 people. These
ships aimed to establish the first European colony in New South Wales. These 11 ships
were named First Fleet.

User:
{Input}
```

(b) Overall-Rewriting

Figure 5: Prompts used for sentence-level simplification generation (from Wu & Arase (2025)).

```
System:
Please return the Penn Treebank-style constituency parse for the following sentence.
Only return the parse tree. Do not return any additional text or explanation.

User:
We can let you go with our cloaking device.

Assistant:
(S
  (NP (PRP We))
  (VP (MD can)
      (VP (VB let)
          (NP (PRP you))
          (VP (VB go)
              (PP (IN with)
                  (NP (PRP$ our) (NN cloaking) (NN device))))))
  (. .))

User:
{Input}
```

Figure 6: Prompts used for parsing

**You will be provided with the following:**
```
    A source sentence.
    Four simplified versions of the source sentence (0, 1, 2, and 3)
    Word alignments between the source and each simplified sentences, structured as:
        words in the source sentence with indices.
        words in the simplified sentence with indices.
        word alignments following the format: sourceIndex_sourceWord-
simplifiedIndex_simplifiedWord
    Penn Treebank-style parses for the source and each simplified sentence.
```

**Your task:** Act as an evaluation system, choose the best and the worst among four
simplified sentences. Analyze each simplified sentence across three aspects:
```
    Lexical: Refer to word alignments.
    Structural: Refer to the parse trees.
    Overall: Consider both lexical and structural aspects.
    Follow the evaluation principles strictly.
```

**Lexical Evaluation Principles:**
```
    Replacing difficult words with easier ones without changing the meaning, or only
slightly changing it. → High reward
    Replacing easy words with even simpler ones without changing the meaning, or only
slightly changing it. → Moderate reward
    Replacing words with more complex ones → High penalty
    Replacing words in a way that significantly changes the original meaning → High
penalty
    Deleting unimportant difficult words → Moderate reward
    Deleting easy words → Moderate penalty
    Deleting important information → High penalty
    Deleting, replacing, or omitting important named entities → High penalty
    Retaining difficult words → Moderate penalty
    Adding new complex words → High penalty
```

**Structural Evaluation Principles:**
```
    Simplifying difficult structures → High reward
    Simplifying easy structures → Moderate reward
    Using more difficult structures → High penalty
    Splitting long sentences → High reward
    Reordering for clarity → High reward
    Edits that do not contribute to simplicity or clarity → Moderate penalty
    Retaining difficult structures → High penalty
```

**Output Instructions:**
```
    For each aspect (Lexical, Structural, Overall), return:
    Aspect: {Lexical / Structural / Overall}, Best: {0, 1, 2 or 3}, Worst: {0, 1, 2 or
3}
    Strictly follow the above format and do not include any extra symbols.
    For the lexical and structural aspect, also provide the analysis of each simplified
sentence during your evaluation.
```

Figure 7: Guidelines for sentence-level simplifications

**Source sentence:** We'll slip you through with our cloaking device.
**0:** With our cloaking devices, we can slip by you.
**1:** We will hide you using our cloaking device.
**2:** We can let you go with our cloaking device.
**3:** We'll cloak you with our device.

**0 alignment:**
    0_we'll 1_slip 2_you 3_through 4_with 5_our 6_cloaking 7_device.
    0_with 1_our 2_cloaking 3_devices, 4_we 5_can 6_slip 7_by 8_you.
    0_we'll-4_we 1_slip-6_slip 3_through-7_by 4_with-0_with 5_our-1_our 6_cloaking-2_cloaking 7_device.-8_you.
**1 alignment:**
    0_we'll 1_slip 2_you 3_through 4_with 5_our 6_cloaking 7_device.
    0_we 1_will 2_hide 3_you 4_using 5_our 6_cloaking 7_device.
    0_we'll-0_we 0_we'll-1_will 1_slip-2_hide 2_you-3_you 4_with-4_using 5_our-5_our 6_cloaking-6_cloaking 7_device.-7_device.
**2 alignment:**
    0_we'll 1_slip 2_you 3_through 4_with 5_our 6_cloaking 7_device.
    0_we 1_can 2_let 3_you 4_go 5_with 6_our 7_cloaking 8_device.
    0_we'll-0_we 1_slip-2_let 2_you-3_you 3_through-4_go 4_with-5_with 5_our-6_our 6_cloaking-7_cloaking 7_device.-8_device.
**3 alignment:**
    0_we'll 1_slip 2_you 3_through 4_with 5_our 6_cloaking 7_device.
    0_we'll 1_cloak 2_you 3_with 4_our 5_device.
    0_we'll-0_we'll 1_slip-1_cloak 2_you-2_you 4_with-3_with 5_our-4_our 7_device.-5_device.

**Source parse:**
    (S \n  (NP (PRP We))\n (VP\n (MD 'll)\n (VP\n (VB slip)\n (NP (PRP you))\n (PP\n (IN through)\n (PP\n (IN with)\n (NP\n (NP (PRP$ our) (NN cloaking) (NN device))))))\n (. .))
**0 parse:**
    (S \n  (PP (IN With) \n (NP (PRP$ our) (NNS cloaking) (NNS devices)))\n (, ,)\n (NP (PRP we))\n (VP (MD can) \n (VP (VB slip) \n (PRT (RP by)) \n (NP (PRP you))))\n (. .))
**1 parse:**
    (S \n  (NP (PRP We)) \n (VP \n (MD will) \n (VP \n (VB hide) \n (NP (PRP you)) \n (PP \n (VBG using) \n (NP \n (PRP$ our) \n (NN cloaking) \n (NN device)))))\n (. .))
**2 parse:**
    (S \n  (NP (PRP We)) (VP (MD can) (VP (VB let) (NP (PRP you)) (VP (VB go) (PP (IN with) (NP (PRP$ our) (NN cloaking) (NN device)))))) (. .))
**3 parse:**
    (S (NP (PRP We)) (VP (MD 'll) (VP (VB cloak) (NP (PRP you)) (PP (IN with) (NP (PRP$ our) (NN device))))) (. .))

(a) Input

**Aspect: Lexical, Best: 1, Worst: 0**
Analysis:
0: 'slip you through'→'slip by you' reverses the core meaning — high penalty. Also pluralizes "device" to "devices" without justification — light penalty for minor factual change. The difficult word 'cloaking' is retained — moderate penalty.
1: 'slip you through'→'hide you' shifts nuance, but becomes simpler and preserves most of the intended meaning — moderate high reward. The difficult word 'cloaking' is retained — moderate penalty.
2: 'slip you through'→'let you go' may imply release rather than passage — light penalty. The difficult word 'cloaking' is retained — moderate penalty.
3: "slip you through"→"cloak you" simplifies verb but loses the specific "cloaking device" label ("our device" is ambiguous) → moderate-high penalty for deleting an important modifier

**Aspect: Structural, Best: 3, Worst: 2**
Analysis:
0: Fronting the PP ("With our…devices") aids clarity. However, it fails to simplify the source's nested PPs; instead, creates a new multi-layered VP (slip → by → you) without resolving recursion.
1: Collapses two nested PPs ("through with...") with a single gerundive phrase ("using our cloaking device"), resulting in a shallower syntactic tree.
2: Avoids the nested PP by using dual VP embedding ("let you go"), but less streamlined than 1: The PP ("with...") modifies the lower verb (go), maintaining a moderately hierarchical structure.
3: Simple S → NP VP PP, no nesting → high reward

**Aspect: Overall, Best: 1, Worst: 0**

(b) Corresponding output showing evaluations on lexical, structural, and overall dimensions.

Figure 8: Example 1 for LLM-as-a-Judge

**Source sentence:** The humans had grown complacent, having faced no concentrated robotic aggression during the last century of uneasy cold war.
0: People were beginning to be satisfied because during the last hundred years of a nervous cold war, they had not had to face any concentrated aggression of robots.
1: The humans became complacent due to a century without robotic aggression.
2: They had become complacent.
3: The humans became complacent, having faced no major robotic attacks in the last century of cold war tensions.

**0 alignment:**
   0_the 1_humans 2_had 3_grown 4_complacent, 5_having 6_faced 7_no 8_concentrated 9_robotic 10_aggression 11_during 12_the 13_last 14_century 15_of 16_uneasy 17_cold 18_war.
   0_people 1_were 2_beginning 3_to 4_be 5_satisfied 6_because 7_during 8_the 9_last 10_hundred 11_years 12_of 13_a 14_nervous 15_cold 16_war, 17_they 18_had 19_not 20_had 21_to 22_face 23_any 24_concentrated 25_aggression 26_of 27_robotes.
   1_humans-0_people 2_had-1_were 3_grown-4_be 4_complacent,-5_satisfied 5_having-18_had 6_faced-22_face 7_no-23_any 8_concentrated-24_concentrated 10_aggression-25_aggression 11_during-7_during 12_the-8_the 13_last-9_last 14_century-11_years 15_of-12_of 16_uneasy-14_nervous 17_cold-15_cold 18_war.-27_robotes.
**1 alignment:**
   0_the 1_humans 2_had 3_grown 4_complacent, 5_having 6_faced 7_no 8_concentrated 9_robotic 10_aggression 11_during 12_the 13_last 14_century 15_of 16_uneasy 17_cold 18_war.
   0_the 1_humans 2_became 3_complacent 4_due 5_to 6_a 7_century 8_without 9_robotic 10_aggression.
   0_the-0_the 1_humans-1_humans 3_grown-2_became 4_complacent,-3_complacent 7_no-8_without 9_robotic-9_robotic 10_aggression-10_aggression. 12_the-6_a 14_century-7_century 18_war.-10_aggression.
**2 alignment:**
   0_the 1_humans 2_had 3_grown 4_complacent, 5_having 6_faced 7_no 8_concentrated 9_robotic 10_aggression 11_during 12_the 13_last 14_century 15_of 16_uneasy 17_cold 18_war.
   0_they 1_had 2_become 3_complacent.
   0_the-0_they 1_humans-0_they 2_had-1_had 3_grown-2_become 4_complacent,-3_complacent. 5_having-1_had 18_war.-3_complacent.
**3 alignment:**
   0_the 1_humans 2_had 3_grown 4_complacent, 5_having 6_faced 7_no 8_concentrated 9_robotic 10_aggression 11_during 12_the 13_last 14_century 15_of 16_uneasy 17_cold 18_war.
   0_the 1_humans 2_became 3_complacent, 4_having 5_faced 6_no 7_major 8_robotic 9_attacks 10_in 11_the 12_last 13_century 14_of 15_cold 16_war 17_tensions.
   0_the-0_the 1_humans-1_humans 3_grown-2_became 4_complacent,-3_complacent, 5_having-4_having 6_faced-5_faced 7_no-6_no 8_concentrated-7_major 9_robotic-8_robotic 10_aggression-9_attacks 11_during-10_in 12_the-11_the 13_last-12_last 14_century-13_century 15_of-14_of 17_cold-15_cold 18_war.-17_tensions.

**Source parse:**
   (S (NP (DT The) (NNS humans)) (VP (VBD had) (VP (VBN grown) (ADJP (JJ complacent), (S (VP (VBG having) (VP (VBN faced) (NP (DT no) (JJ concentrated) (JJ robotic) (NN aggression)) (PP (IN during) (NP (DT the) (JJ last) (NN century) (PP (IN of) (NP (JJ uneasy) (JJ cold) (NN war))))))))))) (. .))
**0 parse:**
   (S (NP (NNS People)) (VP (VBD were) (VP (VBG beginning) (S (VP (TO to) (VP (VB be) (VP (VBN satisfied) (SBAR (IN because) (S (PP (IN during) (NP (DT the) (JJ last) (CD hundred) (NNS years) (PP (IN of) (NP (DT a) (JJ nervous) (NNP Cold) (NNP War))))) (, ,) (NP (PRP they)) (VP (VBD had) (RB not) (VP (VBN had) (S (VP (TO to) (VP (VB face) (NP (DT any) (JJ concentrated) (NN aggression) (PP (IN of) (NP (NNS robots))))))))))))))))) (. .))
**1 parse:**
   (S (NP \n  (NP (DT The) (NNS humans)) (VP (VBD became) (ADJP (JJ complacent) (PP (IN due) (PP (TO to) (NP (DT a) (NN century) (PP (IN without) (NP (JJ robotic) (NN aggression)))))))) (. .))
**2 parse:**
   (S (NP \n  (NP (PRP They)) (VP (VBD had) (VP (VBN become) (ADJP (JJ complacent)))) (. .)
**3 parse:**
   (S (NP (DT The) (NNS humans)) (VP (VBD became) (ADJP (JJ complacent)) (, ,) (S (VP (VBG having) (VP (VBN faced) (NP (DT no) (JJ major) (JJ robotic) (NNS attacks)) (PP (IN in) (NP (DT the) (JJ last) (NN century) (PP (IN of) (NP (JJ cold) (NN war) (NNS tensions))))))))) (. .))

(a) Input

**Aspect: Lexical, Best: 3, Worst: 2**
Analysis:
0: Simplifies "humans" → "people", "century" → "hundred years", "had grown complacent" → "were beginning to be satisfied". High meaning preservation. "robotic aggression"→"aggression of robots", moderate penalty. Adds filler "because," "they had not had to": moderate penalty.
1: Simplifies "had grown" → "became" (reduces verb complexity) → high reward. Omits "concentrated" and "cold war" nuance → moderate penalty. Retains "complacent" (difficult word) → moderate penalty. Good lexical clarity with minor losses.
2: "humans" → "they" (loss of specificity) → high penalty. Retains "complacent" (difficult word) → moderate penalty. Heavy meaning loss despite surface simplicity.
3: "had grown complacent"→"became complacent": moderate reward. Retains "complacent" (difficult word): moderate penalty. "concentrated"→"major," "aggression"→"attacks": simpler synonyms—high reward.

**Aspect: Structural, Best: 2, Worst: 0**
Analysis:
0: Retains complex multi-clause structure: nested VP chains, SBAR ("because"), deeply embedded PPs. Adds a "because" link (small clarity gain), but doesn't fundamentally flatten structure.
1: Flattens source's multi-clause structure into a single VP-ADJP chain. Simplifies "during...century" → "due to a century without" (reduces PP embedding → high reward).
2: Collapses the source into minimal NP-VP-ADJP structure (no clauses or modifiers → high reward).
3: Retains the original complex participial construction ("having faced…"), only substituting simpler words, moderate structural penalty for not simplifying the structure.

**Aspect: Overall, Best: 1, Worst: 2**

(b) Corresponding output showing evaluations on lexical, structural, and overall dimensions.

Figure 9: Example 2 for LLM-as-a-Judge

**Source sentence:** These concerns have intensified due to the actions of China, the predominant supplier.
**0:** These concerns have gotten larger because of the actions of China.
**1:** China's actions have intensified these concerns.
**2:** These are concerns that have been raised by China, who are a major supplier.
**3:** These concerns have intensified because of China's actions, which is the main supplier.

**0 alignment:**
    0_these 1_concerns 2_have 3_intensified 4_due 5_to 6_the 7_actions 8_of 9_china, 10_the 11_predominant 12_supplier.
    0_these 1_concerns 2_have 3_gotten 4_larger 5_because 6_of 7_the 8_actions 9_of 10_china.
    0_these-0_these 1_concerns-1_concerns 2_have-2_have 3_intensified-3_gotten 3_intensified-4_larger 4_due-5_because 5_to-6_of 6_the-7_the 7_actions-8_actions 8_of-9_of 12_supplier.-10_china.
**1 alignment:**
    0_these 1_concerns 2_have 3_intensified 4_due 5_to 6_the 7_actions 8_of 9_china, 10_the 11_predominant 12_supplier.
    0_china's 1_actions 2_have 3_intensified 4_these 5_concerns.
    0_these-4_these 1_concerns-5_concerns. 2_have-2_have 3_intensified-3_intensified 7_actions-1_actions 9_china,-0_china's 12_supplier.-5_concerns.
**2 alignment:**
    0_these 1_concerns 2_have 3_intensified 4_due 5_to 6_the 7_actions 8_of 9_china, 10_the 11_predominant 12_supplier.
    0_these 1_are 2_concerns 3_that 4_have 5_been 6_raised 7_by 8_china, 9_who 10_are 11_a 12_major 13_supplier.
    0_these-0_these 1_concerns-2_concerns 2_have-4_have 3_intensified-6_raised 8_of-7_by 9_china,-8_china, 10_the-11_a 11_predominant-12_major 12_supplier.-13_supplier.
**3 alignment:**
    0_these 1_concerns 2_have 3_intensified 4_due 5_to 6_the 7_actions 8_of 9_china, 10_the 11_predominant 12_supplier.
    0_these 1_concerns 2_have 3_intensified 4_because 5_of 6_china's 7_actions, 8_which 9_is 10_the 11_main 12_supplier.
    0_these-0_these 1_concerns-1_concerns 2_have-2_have 3_intensified-3_intensified 4_due-4_because 5_to-5_of 7_actions-7_actions, 10_the-10_the 11_predominant-11_main 12_supplier.-12_supplier.

**Source parse:**
    (S (NP (DT These) (NNS concerns)) (VP (AUX have) (VP (VBN intensified) (PP (IN due) (PP (TO to) (NP (DT the) (NNS actions) (PP (IN of) (NP (NNP China) (, ,) (NP (DT the) (JJ predominant) (NN supplier))))))))) (. .))
**0 parse:**
    (S \n  (NP (DT These) (NNS concerns)) (VP (VBP have) (VP (VBN gotten) (ADJP (JJR larger)) (SBAR (IN because) (S (PP (IN of) (NP (DT the) (NNS actions))) (PP (IN of) (NP (NNP China)))))))) .)
**1 parse:**
    (S \n  (NP (NNP China) (POS 's))\n  (VP \n    (VBZ actions) \n    (VP \n      (VBP have) \n      (VP \n        (VBN intensified) \n        (NP \n          (DT these) \n          (NNS concerns)))))\n  (. .))
(S (NP (NNP China) (POS 's)) (VP (VBZ actions) (VP (VBP have) (VP (VBN intensified) (NP (DT these) (NNS concerns))))) (. .))
**2 parse:**
    (S (NP (DT These)) (VP (VBP are) (NP (NNS concerns) (SBAR (WHNP (WDT that)) (S (VP (VBP have) (VP (VBN been) (VP (VBN raised) (PP (IN by) (NP (NNP China))) (, ,) (SBAR (WHNP (WP who)) (S (VP (VBP are) (NP (DT a) (JJ major) (NN supplier)))))))))))))
**3 parse:**
    (S (NP (DT These) (NNS concerns)) (VP (VBP have) (VP (VBN intensified) (PP (IN because) (PP (IN of) (NP (NP (NNP China) (POS 's) (NNS actions)) (SBAR (WHNP (WDT which)) (S (VP (VBZ is) (NP (DT the) (JJ main) (NN supplier)))))))))) (. .))

(a) Input

**Aspect: Lexical, Best: 3, Worst: 2**
Analysis:
0: Simplifies "intensified" with "gotten larger, "due to" with "because of" → high reward. Deletes "predominant supplier" - loss of specificity but retains core meaning, moderate penalty.
1: Retains the complex word "intensified", moderate penalty. Deletes "predominant supplier" - loss of specificity but retains core meaning, moderate penalty.
2: "intensified" → "raised" (significant meaning change), high penalty. Simplifies "predominant" -> "major", high reward. However, meaning distortion outweighs its partial simplification.
3: Retains the complex word "intensified", moderate penalty. Simplifies "predominant" with "main", "due to" with "because of", high reward. Keeps most of the original content.

**Aspect: Structural, Best: 1, Worst: 2**
Analysis:
0: Retains complex SBAR ("because of") and nested PPs, adding no structural improvement.
1: Flattens the source's structure into a clean active-voice NP-VP-NP ("China's actions have intensified concerns") → high reward.
2: Introduces two relative clauses ("that have been raised...", "who are..."), creating deeper SBAR nesting than the source → high penalty.
3: Embeds a relative clause ("which is…"), mirroring the source's nesting—moderate penalty.

**Aspect: Overall, Best: 1, Worst: 2**

(b) Corresponding output showing evaluations on lexical, structural, and overall dimensions.

Figure 10: Example 3 for LLM-as-a-Judge

```
You will be provided with the following:
    A source sentence and its simplified version.
    Word alignments between the source and the simplified sentence, structured as:
        words in the source sentence with indices.
        words in the simplified sentence with indices.
        word alignments following the format: sourceIndex_sourceWord-
simplifiedIndex_simplifiedWord

Your task: Act as an evaluation system, evaluate the lexical quality of the simplified
sentence referring to the word alignments and the following lexical evaluation
principles:
    Replacing difficult words with easier ones without changing the meaning, or only
slightly changing it. → High reward
    Replacing easy words with even simpler ones without changing the meaning, or only
slightly changing it. → Moderate reward
    Replacing words with more complex ones → High penalty
    Replacing words in a way that significantly changes the original meaning → High
penalty
    Deleting unimportant difficult words → Moderate reward
    Deleting easy words → Moderate penalty
    Deleting important information → High penalty
    Deleting, replacing, or omitting important named entities → High penalty
    Retaining difficult words → Moderate penalty
    Adding new complex words → High penalty

Please give me an evaluation report.
```

(a) Lexical-Paraphrasing

```
You will be provided with the following:
    A source sentence and its simplified version.
    Word alignments between the source and the simplified sentence, structured as:
        words in the source sentence with indices.
        words in the simplified sentence with indices.
        word alignments following the format: sourceIndex_sourceWord-
simplifiedIndex_simplifiedWord
    Penn Treebank-style parses for the source and each simplified sentence.

Your task: Act as an evaluation system, evaluate the overall quality of the simplified
sentence referring to the word alignments, parses, and the following evaluation
principles.
Lexical Evaluation Principles:
    Replacing difficult words with easier ones without changing the meaning, or only
slightly changing it. → High reward
    Replacing easy words with even simpler ones without changing the meaning, or only
slightly changing it. → Moderate reward
    Replacing words with more complex ones → High penalty
    Replacing words in a way that significantly changes the original meaning → High
penalty
    Deleting unimportant difficult words → Moderate reward
    Deleting easy words → Moderate penalty
    Deleting important information → High penalty
    Deleting, replacing, or omitting important named entities → High penalty
    Retaining difficult words → Moderate penalty
    Adding new complex words → High penalty

Structural Evaluation Principles:
    Simplifying difficult structures → High reward
    Simplifying easy structures → Moderate reward
    Using more difficult structures → High penalty
    Splitting long sentences → High reward
    Reordering for clarity → High reward
    Edits that do not contribute to simplicity or clarity → Moderate penalty
    Retaining difficult structures → High penalty

Please give me an evaluation report.
```

(b) Overall-Rewriting

Figure 11: Guidelines in iterative feedback setting. Contents are from Figure 7, 8, 9, and 10.

**Source sentence:** We'll slip you through with our cloaking device.
**Simplified sentence:** With our cloaking devices, we can slip by you.

**Alignment:**
    0_we'll 1_slip 2_you 3_through 4_with 5_our 6_cloaking 7_device.
    0_with 1_our 2_cloaking 3_devices, 4_we 5_can 6_slip 7_by 8_you.
    0_we'll-4_we 1_slip-6_slip 3_through-7_by 4_with-0_with 5_our-1_our 6_cloaking-
2_cloaking 7_device.-8_you.

(a) Input of Example 1

'slip you through'→'slip by you' reverses the core meaning — high penalty. Also
pluralizes "device" to "devices" without justification — light penalty for minor
factual change. The difficult word 'cloaking' is retained — moderate penalty.

(b) Output of Example 1

**Source sentence:**. These concerns have intensified due to the actions of China, the
predominant supplier.
**Simplified sentence:** These are concerns that have been raised by China, who are a
major supplier.

**Alignment:**
    0_these 1_concerns 2_have 3_intensified 4_due 5_to 6_the 7_actions 8_of 9_china,
10_the 11_predominant 12_supplier.
    0_these 1_are 2_concerns 3_that 4_have 5_been 6_raised 7_by 8_china, 9_who
10_are 11_a 12_major 13_supplier.
    0_these-0_these 1_concerns-2_concerns 2_have-4_have 3_intensified-6_raised
8_of-7_by 9_china,-8_china, 10_the-11_a 11_predominant-12_major 12_supplier.-
13_supplier.

(c) Input of Example 2

"intensified" → "raised" (significant meaning change), high penalty. Simplifies
"predominant" -> "major", high reward. However, meaning distortion outweighs its
partial simplification.

(d) Output of Example 2

**Source sentence:** The humans had grown complacent, having faced no concentrated
robotic aggression during the last century of uneasy cold war.
**Simplified sentence:** The humans became complacent, having faced no major robotic
attacks in the last century of cold war tensions.

**Alignment:**
    0_the 1_humans 2_had 3_grown 4_complacent, 5_having 6_faced 7_no 8_concentrated
9_robotic 10_aggression 11_during 12_the 13_last 14_century 15_of 16_uneasy 17_cold
18_war.
    0_the 1_humans 2_became 3_complacent, 4_having 5_faced 6_no 7_major 8_robotic
9_attacks 10_in 11_the 12_last 13_century 14_of 15_cold 16_war 17_tensions.
    0_the-0_the 1_humans-1_humans 3_grown-2_became 4_complacent,-3_complacent,
5_having-4_having 6_faced-5_faced 7_no-6_no 8_concentrated-7_major 9_robotic-
8_robotic 10_aggression-9_attacks 11_during-10_in 12_the-11_the 13_last-12_last
14_century-13_century 15_of-14_of 17_cold-15_cold 18_war.-17_tensions.

(e) Input of Example 3

"had grown complacent"→"became complacent": moderate reward. Retains "complacent"
(difficult word): moderate penalty. "concentrated"→"major," "aggression"→"attacks":
simpler synonyms—high reward.

(f) Output of Example 3

Figure 12: Three-shot examples of lexical paraphrasing used for LLM-as-a-Judge in the iterative
feedback setting.

**Source sentence:** We'll slip you through with our cloaking device.
**Simplified sentence:** With our cloaking devices, we can slip by you.

**Alignment:**
    0_we'll 1_slip 2_you 3_through 4_with 5_our 6_cloaking 7_device.
    0_with 1_our 2_cloaking 3_devices, 4_we 5_can 6_slip 7_by 8_you.
    0_we'll-4_we 1_slip-6_slip 3_through-7_by 4_with-0_with 5_our-1_our 6_cloaking-2_cloaking 7_device.-8_you.

**Source parse:**
    (S \n  (NP (PRP We))\n (VP\n (MD 'll)\n (VP\n (VB slip)\n (NP (PRP you))\n (PP\n (IN through)\n (PP\n (IN with)\n (NP\n (NP (PRP$ our) (NN cloaking) (NN device))))))\n (. .))
**Simplified parse:**
    (S \n  (PP (IN With) \n (NP (PRP$ our) (NNS cloaking) (NNS devices)))\n (, ,)\n (NP (PRP we))\n (VP (MD can) \n (VP (VB slip) \n (PRT (RP by)) \n (NP (PRP you))))\n (. .))

### (a) Input of Example 1

**Lexical Analysis:**
'slip you through'→'slip by you' reverses the core meaning — high penalty. Also pluralizes "device" to "devices" without justification — light penalty for minor factual change. The difficult word 'cloaking' is retained — moderate penalty.

**Structural Analysis:**
Fronting the PP ("With our…devices") aids clarity. However, it fails to simplify the source's nested PPs; instead, creates a new multi-layered VP (slip → by → you) without resolving recursion.

### (b) Output of Example 1

**Source sentence:** The humans had grown complacent, having faced no concentrated robotic aggression during the last century of uneasy cold war.
**Simplified sentence:** The humans became complacent, having faced no major robotic attacks in the last century of cold war tensions.

**Alignment:**
    0_the 1_humans 2_had 3_grown 4_complacent, 5_having 6_faced 7_no 8_concentrated 9_robotic 10_aggression 11_during 12_the 13_last 14_century 15_of 16_uneasy 17_cold 18_war.
    0_the 1_humans 2_became 3_complacent, 4_having 5_faced 6_no 7_major 8_robotic 9_attacks 10_in 11_the 12_last 13_century 14_of 15_cold 16_war 17_tensions.
    0_the-0_the 1_humans-1_humans 3_grown-2_became 4_complacent,-3_complacent, 5_having-4_having 6_faced-5_faced 7_no-6_no 8_concentrated-7_major 9_robotic-8_robotic 10_aggression-9_attacks 11_during-10_in 12_the-11_the 13_last-12_last 14_century-13_century 15_of-14_of 17_cold-15_cold 18_war.-17_tensions.

**Source parse:**
    (S (NP (DT The) (NNS humans)) (VP (VBD had) (VP (VBN grown) (ADJP (JJ complacent), (S (VP (VBG having) (VP (VBN faced) (NP (DT no) (JJ concentrated) (JJ robotic) (NN aggression)) (PP (IN during) (NP (DT the) (JJ last) (NN century) (PP (IN of) (NP (JJ uneasy) (JJ cold) (NN war)))))))))))) (. .))
**Simplified parse:**
    (S (NP (DT The) (NNS humans)) (VP (VBD became) (ADJP (JJ complacent)) (, ,) (S (VP (VBG having) (VP (VBN faced) (NP (DT no) (JJ major) (JJ robotic) (NNS attacks)) (PP (IN in) (NP (DT the) (JJ last) (NN century) (PP (IN of) (NP (JJ cold) (NN war) (NNS tensions)))))))))) (. .))

### (c) Input of Example 2

**Lexical Analysis:**
"had grown complacent"→"became complacent": moderate reward. Retains "complacent" (difficult word): moderate penalty. "concentrated"→"major," "aggression"→"attacks": simpler synonyms–high reward.

**Structural Analysis:**
Retains the original complex participial construction ("having faced…"), only substituting simpler words, moderate structural penalty for not simplifying the structure.

### (d) Output of Example 2

**Source sentence:.** These concerns have intensified due to the actions of China, the predominant supplier.
**Simplified sentence:** These are concerns that have been raised by China, who are a major supplier.

**Alignment:**
    0_these 1_concerns 2_have 3_intensified 4_due 5_to 6_the 7_actions 8_of 9_china, 10_the 11_predominant 12_supplier.
    0_these 1_are 2_concerns 3_that 4_have 5_been 6_raised 7_by 8_china, 9_who 10_are 11_a 12_major 13_supplier.
    0_these-0_these 1_concerns-2_concerns 2_have-4_have 3_intensified-6_raised 8_of-7_by 9_china,-8_china, 10_the-11_a 11_predominant-12_major 12_supplier.-13_supplier.

**Source parse:**
    (S (NP (DT These) (NNS concerns)) (VP (AUX have) (VP (VBN intensified) (PP (IN due) (PP (TO to) (NP (DT the) (NNS actions) (PP (IN of) (NP (NNP China) (, ,) (NP (DT the) (JJ predominant) (NN supplier)))))))))) (. .))
**Simplified parse:**
    (S (NP (DT These)) (VP (VBP are) (NP (NNS concerns) (SBAR (WHNP (WDT that)) (S (VP (VBP have) (VP (VBN been) (VP (VBN raised) (PP (IN by) (NP (NNP China))) (, ,) (SBAR (WHNP (WP who)) (S (VP (VBP are) (NP (DT a) (JJ major) (NN supplier)))))))))))))

### (e) Input of Example 3

**Lexical Analysis:**
"intensified" → "raised" (significant meaning change), high penalty. Simplifies "predominant" -> "major", high reward. However, meaning distortion outweighs its partial simplification.

**Structural Analysis:**
Introduces two relative clauses ("that have been raised...", "who are..."), creating deeper SBAR nesting than the source → high penalty.

### (f) Output of Example 3

Figure 13: 3-shot examples of overall paraphrasing used for LLM-as-a-Judge in the iterative feedback setting

```
System:
You will be provided a source sentence, a simplified sentence, and an evaluation report
assessing the simplification. Improve the simplified sentence according to the
evaluation. If there are no issues mentioned in the evaluation, return the original
simplified sentence. Return only the revised simplified sentence without numbering or
explanations.

User:
{Input}
```

Figure 14: Prompts used for self-correction in the iterative feedback setting

```
You are required to simplify documents while preserving the meaning, maintaining
coherence, and improving cohesion. Return only the simplified document without
numbering or explanations.
```

Figure 15: Prompts used for document-level simplification generation

```
You will be provided with the following:
   A source document and four simplified versions of the source document (0, 1, 2, and 3)

Your task: Act as an evaluation system, choose the best and the worst among four simplified documents.
Analyze each simplified sentence across four aspects: simplicity, discourse relations, coreference,
global coherence.

Simplicity Evaluation Principles:
   Replace complex words/phrases with simpler but accurate ones → High reward
   Split very long sentences into shorter and simpler ones → High reward
   Remove minor redundancies (repeated or obvious info) → Moderate reward
   Keep complex words/phrases unchanged when they could be simplified. → Moderate penalty
   Delete important information → High penalty
   Replace precise terms with overly vague or wrong simple words/phrases → High penalty

Discourse Relations Evaluation Principles:
   Add or keep discourse markers that clarify relationships between sentences/clauses → High reward
   Reorder sentences/clauses to improve clarity of discourse relations → Moderate reward
   Using more difficult structures → High penalty
   Delete discourse markers that clarify relationships between sentences/clauses → High penalty

Coreference Evaluation Principles:
   Keep pronouns when antecedent is clear and close → Moderate reward
   Use pronouns with ambiguous antecedents → High penalty
   Change a referring expression so it points to the wrong entity → High penalty
   Replace unclear pronouns with explicit, correct noun phrases → Moderate reward

Global Coherence Evaluation Principles:
   Keep or improve a clear high-level order and strong global coherence → High reward
   Scatter logically related info far apart without signals → High penalty
   Delete info crucial for overall understanding → High penalty

Return your answer in the following format:
   Best: {0, 1, 2 or 3}, Worst: {0, 1, 2 or 3}
Strictly follow the above format and do not include any explanation or extra symbols.
```

Figure 16: Guidelines for document-level simplifications

