# OpenReview forum: "Policy-Based Sentence Simplification: Replacing Parallel Corpora with LLM-as-a-Judge"
_ICLR.cc/2026/Conference — Submitted to ICLR 2026_

### Official Review · Reviewer_DkvM · 2025-10-16

**Soundness:** 3
**Presentation:** 4
**Contribution:** 2
**Rating:** 4
**Confidence:** 4

**Summary:**

The paper proposes a method to control the quality of type of edits in sentence simplification without relying on human-annotated corpora. The approach is to use LLMs both generate candidate simplifications and to judge pairwise preferences, then train a decoder‑only policy model with preference optimization (ARPO). The authors design prompt-based guidelines for rewards/penalties for specific operations, where OTAlign is used as lexical judge and Qwen is used as structural judge.

**Strengths:**

1. The paper is well written and easy to follow.
2. The overall pipeline is conceptually simple and can be adapted to different edit policies and deployment scenarios.
3. Experiments consider multiple baselines and include both automatic and human assessments.
4. Aligning small, open‑source LLMs with this policy framework can match or exceed closed-source LLMs on both automatic and human metrics.

**Weaknesses:**

1. The scope is limited to sentence-level simplification, which I think underutilizes current LLM capabilities. Meanwhile, real‑world needs often require paragraph or document‑level simplification with cross‑sentence constraints and dependencies.


2. The system’s practical value remains unconvincing to me. The approach still focuses on generic sentence‑level edit operations, whereas special groups of users, e.g., people with reading impairments, need solutions tailored to their specific difficulties. A user‑focused design with explicit needs analysis would strengthen the contribution.

3. The desirability of the simplified sentences is judged by LLMs, which is unclear to me as to how it benefits human users or human preference can be effectively taken into account.

**Questions:**

1. What are the prompts for the vanilla base models? (I could not find them reported in the Appendix). How does the proposed policy compare directly to carefully engineered prompts (with iterative feedback) that request specific edit types, both in quality and controllability?

2. What is the performance on out‑of‑domain sentences (e.g., different genres, technical vs. news)?  Is retraining required, or can the policy adapt via light tuning in these scenarios?

3. Can the framework be extended to paragraph/document‑level simplification? What changes are needed to handle discourse relations, coreference, and global coherence?

---

> ### Author Response · Authors · 2025-11-21
>
> Thank you for recognizing the contribution of our work. We address the concerns raised by the reviewer below.
>
> **Comment 1:**
> The scope is limited to sentence-level simplification, which I think underutilizes current LLM capabilities. Meanwhile, real-world needs often require paragraph or document-level simplification with cross-sentence constraints and dependencies. Can the framework be extended to paragraph/document-level simplification? What changes are needed to handle discourse relations, coreference, and global coherence?
>
> **Response:**
> Thank you for the thoughtful comment. Our study focuses on sentence-level simplification, where LLMs remain error-prone and struggle to consistently align with human preferences (Heinemann et al., 2023; Barayan et al., 2025). Furthermore, we would remark that sentence-level simplification is desirable in a language education context (which is our focus) because teachers can easily track where changes have been made. We have received feedback from language teachers that such tracability is crucial in educational applications.
>
> Having said that, we agree that paragraph- and document-level simplification is also an important direction. We conducted preliminary experiments indicating that our framework can be naturally extended to document level, with only modest changes. The overall three-step pipeline remains unchanged; the necessary adaptations involve adjusting the LLM-as-a-Judge guidelines to incorporate those aspects addressed by the reviewer. The following summarizes our experiment setup and results.
>
> ---
>
> **Training**
>
> - **Step 1: Candidate Pool for Preference Data**
> We used around 1.8k complex documents from the Newsela dataset as inputs (Xu, 2015), and collected simplifications from the same group of LLMs (see Figure 1). Importantly, we did not include Newsela reference simplifications in the candidate pool.
> - **Step 2: LLM-as-a-Judge**
> To address the reviewer’s concerns directly, we expanded the evaluation guidelines to cover: simplicity, discourse relations, coreference and global coherence. The evaluation principles include edit types, their expected effects, and associated rewards or penalties. Based on these principles, LLM-as-a-Judge selected preferred candidates and dispreferred candidates.
> - **Step 3: Preference Optimization**
> We trained models (Qwen2.5-7B and Llama3.1-8B) with the resulting preference dataset following the same preference-optimization procedure as in the paper.
>
> ---
>
>  **Evaluation Setup**
>
> We used 200 held-out Newsela documents and their simplification references. Two automatic metrics, Agg-LENS and Agg-BERTScore (Maddela & Alva-Manchego, 2025) were used for evaluation:
> - **Agg-LENS**: the state-of-the-art document-level simplification metric with the strongest correlation to human judgments.
> - **Agg-BERTScore**, which shows the highest consistency for coherence evaluation in simplification.
>
> ---
>
> **Results**
>
> Compared to zero-shot (zs) and few-shot (fs) prompting of vanilla models, preference-optimized models show substantial improvements:
>
>  **Qwen2.5-7B**
>
> | Method-Prompt         | Agg-LENS | Agg-BERT |
> |-----------------------|----------|----------|
> | vanilla-zs            | 37.1     | 15.1     |
> | vanilla-fs            | 39.5     | 17.0     |
> | po_think-zs (Ours) | **46.2** | **25.0** |
>
> **Llama3.1-8B**
>
> | Method-Prompt         | Agg-LENS | Agg-BERT |
> |-----------------------|----------|----------|
> | vanilla-zs            | 36.0     | 16.0     |
> | vanilla-fs            | 43.5     | 17.4     |
> | po_think-zs (Ours) | **48.0** | **27.7** |
>
> However, due to the lack of diverse, policy-aligned test sets and corresponding evaluation metrics for document-level simplification, we were unable to further explore different policies. We will add these results and discussion to the future work section.
>
> ---
>
> **References**
>
> [1] David Heineman, Yao Dou, Mounica Maddela, and Wei Xu. *Dancing between success and failure: Edit-level simplification evaluation using SALSA.* In Proceedings of the 2023 Conference on Empirical Methods in Natural Language Processing, pp. 3466–3495, December 2023.
>
> [2] Abdullah Barayan, Jose Camacho-Collados, and Fernando Alva-Manchego. *Analysing zero-shot readability-controlled sentence simplification.* Proceedings of the 31st International Conference on Computational Linguistics, pp. 6762–6781, Abu Dhabi, UAE, January 2025.
>
> [3] Wei Xu, Chris Callison-Burch, and Courtney Napoles. *Problems in current text simplification research: New data can help.* Transactions of the Association for Computational Linguistics, 3:283–297, 2015.
>
> [4] Mounica Maddela and Fernando Alva-Manchego. *Adapting sentence-level automatic metrics for document-level simplification evaluation.* In Proceedings of the 2025 Conference of the Nations of the Americas Chapter of the Association for Computational Linguistics: Human Language Technologies (Volume 1: Long Papers), pp. 6444–6459, Albuquerque, New Mexico, April 2025.

---

> ### Author Response · Authors · 2025-11-21
>
> **Comment 2:**
> The approach still focuses on generic sentence‑level edit operations, whereas special groups of users, e.g., people with reading impairments, need solutions tailored to their specific difficulties. A user‑focused design with explicit needs analysis would strengthen the contribution.
>
> **Response:**
>
> We appreciate the reviewer’s insightful comment. We respectfully point out that our study is not a “generic” simplification; rather, it is explicitly designed for **second language  (L2) learners**, our target user group. Previous studies have shown that the desired type and degree of simplification edits change depending on learner proficiency and readability levels (Agrawal et al., 2021; Zhong, 2020). Specifically, **low- to intermediate-level learners** benefit from a combination of lexical paraphrasing, structural modifications, and selective deletions to reduce cognitive load.  In contrast, **advanced learners** benefit from lexical paraphrasing, which supports vocabulary acquisition (Chen, 2019), but they gain comparatively less from added cohesion or deletion (Hosoda, 2016; Zhong, 2020).
>
> Motivated by these findings from both the NLP and language education literature, we designed two distinct policies—**Overall-Rewriting** for low- to intermediate-level learners and **Lexical-Paraphrasing** for advanced learners—as described in Introduction (first paragraph) and illustrated in Table 1.  We will make this design motivation clearer in the revision. We fully agree that users with reading impairments or other specific needs represent important target groups. However, it falls outside the scope of the current work.
>
> References
>
> [1] Sweta Agrawal, Weijia Xu, and Marine Carpuat. A non-autoregressive edit-based approach to controllable text simplification. In Findings of the Association for Computational Linguistics: ACL-IJCNLP 2021, pp. 3757–3769, Online, August 2021.
>
> [2] Yang Zhong, Chao Jiang, Wei Xu, and Junyi Jessy Li. Discourse level factors for sentence deletion in text simplification. In Proceedings of the AAAI conference on artificial intelligence, volume 34, pp. 9709–9716, 2020.
>
> [3] Mei-Hua Chen. Phrasal paraphrase learning: Exploring an effective strategy to consolidate vocabulary knowledge. Taiwan Journal of TESOL, 16(1):41–66, 2019.
>
> [4] Masaya Hosoda. The interplay of text cohesion and L2 reading proficiency in different levels of text comprehension among EFL readers. ARELE: Annual Review of English Language Education in Japan, 27:201–216, 2016.
>
> ---
>
> **Comment 3:**
>
> The desirability of the simplified sentences is judged by LLMs, which is unclear to me as to how it benefits human users or human preference can be effectively taken into account.
>
> **Response:**
>
> Thanks for your comments. As clarified in our response to Comment 2, our method is designed for L2 learners, and our edit policies are grounded in findings from both the NLP and language education literature. Previous studies identify what kinds of edits are desirable for learners at different proficiency levels, and we translate these insights into explicit judging guidelines for LLM-as-a-Judge (covering preferred edit types, quality criteria, and reward/penalty rules). In this way, the LLM judge operationalizes human-derived standards rather than relying on arbitrary model judgments. Our human evaluation results (Table 4) further confirm that the simplifications by our models are indeed preferred by human annotators, demonstrating that the LLM-based judging process produces outputs aligned with real user preferences.

---

> ### Author Response · Authors · 2025-11-21
>
> **Comment 4:**
>
> What are the prompts for the vanilla base models? (I could not find them reported in the Appendix). How does the proposed policy compare directly to carefully engineered prompts (with iterative feedback) that request specific edit types, both in quality and controllability?
>
> **Response:**
>
> Thanks for your suggestion. As noted in Appendix A.2.2, we used identical zero-shot prompts for all vanilla and fine-tuned models for comparison. This is because we observed that directly using the few-shot in Figure 5 sometimes led to a performance drop compared to zero-shot, for both vanilla and fine-tuned models. For example, the LENS score of the vanilla Qwen2.5-14B model on the ASSET dataset decreased by 1.5 points, and the SARI score of the Qwen2.5-7B model fine-tuned on parallel corpora decreased by 0.3 points. We conjecture that this is due to the bias introduced by few-shot examples; we therefore adopted the zero-shot setting.
>
> We agree that a comparison to carefully engineered prompts with iterative feedback would further strengthen the paper. As human feedback is too costly to employ in practical applications, we conducted an additional experiment using LLM-as-a-Judge and self-correction: we used the same Judge model and evaluation guidelines to provide iterative feedback to the LLM, and requested the model to do self-correction for two rounds (Iter 1 & Iter 2).  Results show that our method outperforms iterative feedback across two policies. Moreover, iterative feedback resulted in performance degradation in Overall-Rewriting. After taking a closer look, we found that the models tend to become more conservative once they receive feedback, which leads to fewer effective simplification edits.
>
> **Overall-Rewriting: ASSET-LENS**
>
> | Method | Phi3.8B | Qwen7B | Llama8B | Qwen14B |
> |--------|---------|--------|---------|----------|
> | Vanilla | 68.9 | 68.7 | 65.6 | 68.1 |
> | Iter 1  | 66.6 | 63.7 | 64.3 | 64.8 |
> | Iter 2  | 66.0 | 63.6 | 64.0 | 65.7 |
> | Ours | **71.6** | **70.2** | **69.9** | **69.9** |
>
> **Lexical-Paraphrasing: Turk-SARI**
>
> | Method | Phi3.8B | Qwen7B | Llama8B | Qwen14B |
> |--------|---------|--------|---------|----------|
> | Vanilla | 35.3 | 37.9 | 39.1 | 38.5 |
> | Iter 1  | 38.9 | 39.7 | 42.4 | 41.8 |
> | Iter 2  | 39.8 | 40.6 | 42.8 | 42.3 |
> | Ours | **43.3** | **43.7** | **43.7** | **43.9** |
>
> We will include these results in the revised version.
>
> ---
>
> **Comment 5:**
>
> What is the performance on out‑of‑domain sentences (e.g., different genres, technical vs. news)? Is retraining required, or can the policy adapt via light tuning in these scenarios?
>
> **Response:**
>
> Thank you for raising this point. To examine out-of-domain generalization, we evaluated our method on two datasets outside the Wikipedia domain of ASSET and Turk: **SimPA** in the public administration domain (Scarton, 2018),  and **Newsela** in the news domain (Xu, 2015). We directly used our policy models with the same prompts and hyperparameters as in the Wiki-domain experiments, without any retraining. SimPA provides parallel references for both overall rewriting and lexical paraphrasing, so we evaluated both of our policy models on this dataset. Newsela contains simplifications that involve diverse editing operations, so we used our overall rewriting policy models there.
>
> In both domains, our models outperform the vanilla baselines, indicating that the learned policies can transfer across domains. Belows are the results structured as {Dataset} - {Policy} - {Metric}:
>
> **SimPA – Overall-Rewriting – LENS**
>
> | Method  | Phi3.8B | Qwen7B | Llama8B | Qwen14B |
> |---------|---------|--------|---------|---------|
> | Vanilla | 60.1    | 56.0   | 59.8    | 58.7    |
> | Ours    | **63.0**    | **62.7**   | **61.3**    | **61.6**    |
> [Note: GPT4o -> 59.7]
>
> **SimPA – Lexical-Paraphrasing – SARI**
>
> | Method  | Phi3.8B | Qwen7B | Llama8B | Qwen14B |
> |---------|---------|--------|---------|---------|
> | Vanilla | 22.3    | 26.5   | 27.7    | 27.5    |
> | Ours    | **37.3**    | **36.4**   | **36.6**    | **36.8**    |
> [Note: GPT4o -> 28.6]
>
> **Newsela – Overall-Rewriting – LENS**
>
> | Method  | Phi3.8B | Qwen7B | Llama8B | Qwen14B |
> |---------|---------|--------|---------|---------|
> | Vanilla | 62.6    | 59.1   | 61.0    | 62.3    |
> | Ours    | **64.9**    | **63.9**   | **63.2**    | **63.6**    |
> [Note: GPT4o -> 60.9]
>
> We will include these results in the revised version.
>
> References
>
> [1] Carolina Scarton, Gustavo Paetzold, and Lucia Specia. SimPA: A sentence-level simplifica-
> tion corpus for the public administration domain. In Proceedings of the Eleventh Interna-
> tional Conference on Language Resources and Evaluation (LREC 2018), May 2018.
>
> [2] Wei Xu, Chris Callison-Burch, and Courtney Napoles. Problems in current text simplification research: New data can help. Transactions of the Association for Computational Linguistics, 3:283–297, 2015. doi: 10.1162/tacl a 00139.

---

> ### Comment · Reviewer_DkvM · 2025-11-25
>
> Thank you for your responses. I also highly appreciate the authors' effort in running the additional experiments.
>
> The paper is strongly motivated and well supported by insights from language and education literature. However, with today's landscape with better LLMs released every few months, the bar is indeed pretty high for writing tasks. English sentence simplification, especially for users of special needs, is a rather niche problem. I think the community would expect English document-level simplification at the very least, and often more complicated setting like multiple domains, languages or multi-modal inputs.
>
> If the system can be extended to handle beyond sentence-level simplification with ease, as in your responses, I strongly recommend the authors to focus on this aspect, conduct more extensive experiments and improve your contribution. I believe it would then be more convincing.
>
> For the selected problem itself, the paper has indeed provided a very solid and comprehensive solution to it. However, I think the paper, with the current contributions, would be a better fit to NLP conferences with specific related tracks. Considering the competitive nature of ML conferences like ICLR, the contribution may not be considered substantial enough by the ML community.
>
> One final comment: while the reported empirical results are convincing (showing that the method is effective), to me the difference in automatic evaluation performance would not matter much since it in fact comes down to whether the simplified texts improve understandability. A method can score a bit lower on automatic metrics but still yield equally readable texts.
>
> For the above reasons, I retain my rating. I hope this does not discourage the authors from improving the work. I'd like to stress again that this does not indicate that your work is not sound. I just personally expect contributions to more significant problem settings.

---

> > ### Author Response · Authors · 2025-11-29
> >
> > We respectfully disagree with the view that sentence-level simplification for learners with special needs is a “niche” or “solved” problem.
> > - **Reason 1: Simplification is not a “one-size-fits-all” problem.**
> >
> > >A substantial body of work emphasizes that simplification applications designed aligning with users’ needs (e.g., Ref. [1]), particularly in second-language education, where tailored simplification remains a highly valued pedagogical tool (e.g., Ref. [2]). In fact, simplifications for learners at different proficiency levels differ significantly (Ref. [3]). Such divergent requirements cannot, and should not, be addressed through a single simplification strategy.
> >
> > - **Reason 2: Document-level simplification requires robust sentence-level operations.**
> >
> > >Even for general sentence simplification, LLMs do not perform flawlessly. Previous studies on both open- and closed-source simplifications show that LLMs remain error-prone (e.g., Ref [4]). For example, LLMs often substitute original words with more difficult alternatives. This type of issue can severely degrade user experience. To build a robust sentence-level system, we incorporate LLM-as-a-Judge at a fine-grained level, including word-level alignment and corresponding edit-level evaluation guidelines.
> >
> > >As we replied under comment 1, while our method is promising to be adapted on the document level, we were unable to further explore the same controlled setting due to the lack of datasets and metrics. As document-level simplification inherently relies on sentence-level simplification, we believe that establishing a reliable and well-grounded sentence-level framework is both necessary and beneficial to the research community and downstream applications.
> >
> > For your last comment, our conclusions do not rely solely on automatic metrics. Human evaluation results further support the effectiveness and practical value of our method.
> >
> > For these reasons, we believe the task setting is significant, unsolved, and aligned with the scope of ICLR.
> >
> > References
> >
> > [1] Isabel Espinosa-Zaragoza, José Abreu-Salas, Elena Lloret, Paloma Moreda, and Manuel Palomar. 2023. A Review of Research-Based Automatic Text Simplification Tools. In Proceedings of the 14th International Conference on Recent Advances in Natural Language Processing, pages 321–330, Varna, Bulgaria. INCOMA Ltd., Shoumen, Bulgaria.
> >
> > [2] Guanlin Li, Yuki Arase, and Noel Crespi. 2025. Aligning Sentence Simplification with ESL Learner’s Proficiency for Language Acquisition. In Proceedings of the 2025 Conference of the Nations of the Americas Chapter of the Association for Computational Linguistics: Human Language Technologies (Volume 1: Long Papers), pages 492–507, Albuquerque, New Mexico. Association for Computational Linguistics.
> >
> > [3] Scott A. Crossley, David Allen, and Danielle S. McNamara. (2011). Text simplification and comprehensible input: A case for an intuitive approach. Language Teaching Research, 16(1), 89-108. https://doi.org/10.1177/1362168811423456
> >
> > [4] Xuanxin WU and Yuki Arase. 2025. An In-depth Evaluation of Large Language Models in Sentence Simplification with Error-based Human Assessment. ACM Trans. Intell. Syst. Technol. Just Accepted (June 2025). https://doi.org/10.1145/3744744

---

### Official Review · Reviewer_RRWX · 2025-10-25

**Soundness:** 2
**Presentation:** 3
**Contribution:** 2
**Rating:** 2
**Confidence:** 3

**Summary:**

In this manuscript, the authors aim to address the challenges of policy-driven control in sentence simplification, including the lack of policy-specific parallel corpora and the poor policy alignment of small-scale open-source LLMs. The proposed LLM-as-a-Judge automatically constructs policy-aligned preference data and uses preference optimization to fine-tune open-source LLMs. Experimental results across automatic metrics and human evaluations show that the method outperforms baselines.

**Strengths:**

In this manuscript, the authors aim to address the challenges of policy-driven control in sentence simplification, including the lack of policy-specific parallel corpora and the poor policy alignment of small-scale open-source LLMs. The proposed LLM-as-a-Judge automatically constructs policy-aligned preference data and uses preference optimization to fine-tune open-source LLMs. Experimental results across automatic metrics and human evaluations show that the method outperforms baselines. Notably, small-scale open-source LLMs (e.g., Phi-3-mini-3.8B) fine-tuned via this framework surpass GPT-4o in lexical-paraphrasing and achieve comparable performance in overall-rewriting, verifying the approach’s effectiveness and robustness.

**Weaknesses:**

There are some concerns for the manuscript as follows:
1.The proposed LLM-as-a-Judge select high-quality preference data between overall-rewriting and lexical-paraphrasing policies. It means that the candidate simplifications will be generated by various LLMs, so what is the computational cost of this method? This is an important issue, but it is not discussed in the experiments.
2.The proposed LLM-as-a-Judge selected preference data and fine-tune open source LLMs, the important preference optimization is an existing method. Overall, the innovation of the proposed method is limited.
3.In Section 2, how to fine-tune open-source llms is not mentioned.
4.The selection for baseline methods are inappropriate. The sota sentence simplification have not been introduced.

**Questions:**

See the Weaknesses.

---

> ### Author Response · Authors · 2025-11-21
>
> We thank the reviewer for recognizing the effectiveness of our method. We address the concerns raised by the reviewer below.
>
> **Comment 1:**
>
> The proposed LLM-as-a-Judge select high-quality preference data between overall-rewriting and lexical-paraphrasing policies. It means that the candidate simplifications will be generated by various LLMs, so what is the computational cost of this method?
>
> **Response:**
>
> Thanks for your question. Both the simplification generation and the LLM-as-a-Judge selection only need to be done once in our setup.  Generating 8k candidate simplifications took about 3 minutes per LLM on a NVIDIA A6000 Ada 48GB (we used four LLMs; software details are provided in Appendix A.1). The LLM-as-a-Judge step, which uses the large-reasoning model Qwen3-32B, required about 7 hours to produce judgments for all 8k samples on a NVIDIA H100 SXM5 94GB GPU. We believe this cost is reasonable for an offline data-construction pipeline, and we will include these details in the revised version.
>
> ---
>
> **Comment 2:**
>
> The proposed LLM-as-a-Judge selected preference data and fine-tune open source LLMs, the important preference optimization is an existing method. Overall, the innovation of the proposed method is limited.
>
> **Response:**
>
> We believe that showing how to achieve policy-aligned simplifications without relying on parallel corpus, can be a recognisable contribution. And preference learning algorithm is one component in our framework in order to achieve this goal.
>
> As discussed in the introduction and Section 3, sentence simplification poses unique challenges: it requires different simplification policies for language learners with different proficiency levels; however, LLMs are insensitive to policy adaptation via prompting. To address these constraints, we carefully crafted guidelines and incorporated external resources (e.g., alignment and parsing information) to direct LLMs to produce reliable judgments. These judgments then allow us to construct a high-quality preference dataset, which we use to train different LLMs in a simple yet effective manner.
>
> ---
>
> **Comment 3:**
>
> In Section 2, how to fine-tune open-source llms is not mentioned.
>
> **Response:**
>
> We respectfully note that the details of our fine-tuning method are clearly described in  Section 4.3 Comparison Methods.
>
> ---
>
> **Comment 4:**
>
> The selection for baseline methods are inappropriate. The sota sentence simplification have not been introduced.
>
> **Response:**
>
> Thanks for your comment. Previous studies have shown that ChatGPT achieved SOTA performance in sentence simplification (Kew et al., 2023; Wu & Arase, 2025) . We followed these findings and therefore selected ChatGPT models as our upper-bound baselines. Because the reviewer did not cite specific papers, we are unsure which methods are being considered as the state of the art and thus required for comparison. If the reviewer has specific recommendations that outperform ChatGPT on the TURK and ASSET datasets, we would be happy to include them in the revised version.
>
> References
>
> [1] Tannon Kew, Alison Chi, Laura Vásquez-Rodríguez, Sweta Agrawal, Dennis Aumiller, Fernando Alva-Manchego, and Matthew Shardlow. 2023. BLESS: Benchmarking Large Language Models on Sentence Simplification. In Proceedings of the 2023 Conference on Empirical Methods in Natural Language Processing, pages 13291–13309, Singapore. Association for Computational Linguistics.
>
> [2] Xuanxin WU and Yuki Arase. 2025. An In-depth Evaluation of Large Language Models in Sentence Simplification with Error-based Human Assessment. ACM Trans. Intell. Syst. Technol. Just Accepted (June 2025). https://doi.org/10.1145/3744744

---

### Official Review · Reviewer_hfEB · 2025-10-31

**Soundness:** 2
**Presentation:** 3
**Contribution:** 2
**Rating:** 4
**Confidence:** 5

**Summary:**

The paper focus on training small LLMs doing sentence simplification. The method involves sampling simplification outputs from different models and then use LLM as judge to construct preference pairs, and then apply preference learning algorithm.

**Strengths:**

The paper is clear

**Weaknesses:**

The whole pipeline of constructing preference pairs and apply the preference learning algorithms has been used in the past 2 years since the RLHF came out. For example, "Self-Rewarding Language Models" ICML 2024. And the paper just apply it onto a specific task.

The paper use the GPT-4o as an upper bound, but it is a quite old model, should have the evaluation of the current best propreitary model (GPT-5)'s performance on sentence simplification, which is likely a solved task. And GPT-5 is also cheaper than GPT-4o.

**Questions:**

See weaknesses.

---

> ### Author Response · Authors · 2025-11-21
>
> We address the reviewers' concerns below.
>
> **Comment 1:**
>
> The whole pipeline of constructing preference pairs and apply the preference learning algorithms has been used.[...] And the paper just apply it onto a specific task.
>
> **Response:**
>
> We believe that showing how to achieve policy-aligned simplifications without relying on parallel corpus, can be a recognisable contribution. And preference learning algorithm is one component in our framework in order to achieve this goal.
>
> As discussed in the introduction and Section 3, sentence simplification poses unique challenges: it requires different simplification policies for language learners with different proficiency levels; however, LLMs are insensitive to policy adaptation via prompting. To address these constraints, we carefully crafted guidelines and incorporated external resources (e.g., alignment and parsing information) to direct LLMs to produce reliable judgments. These judgments then allow us to construct a high-quality preference dataset, which we use to train different LLMs in a simple yet effective manner.
>
> ---
>
> **Comment 2:**
>
> Use GPT-5 as the upper bound instead of GPT-4o, which is likely a solved task.
>
> **Response:**
>
> We respectfully clarify that text simplification remains an open challenge, even with the capabilities of GPT-5. In practice, GPT-4o and GPT-5 show very similar performance on our task. Below is a comparison using the same prompts as in our study, and we observe only negligible differences. All of our models surpass GPT-5 under both policies (see Figures 2 and 4 for more details):
>
>     {Dataset}-{Metric}: GPT-4o {Score} → GPT-5 {Score}
>     Turk-SARI: GPT-4o 41.0 → GPT-5 41.2
>     ASSET-LENS: GPT-4o 68.0 → GPT-5 68.1
>     ASSET-SARI: GPT-4o 47.5 → GPT-5 46.6
>
> As we agree that GPT-5 is a cheaper option, we will include this information in the revised version of the paper.

---

### Official Review · Reviewer_UcVC · 2025-11-02

**Soundness:** 4
**Presentation:** 3
**Contribution:** 4
**Rating:** 8
**Confidence:** 4

**Summary:**

The authors describe an approach for adapting LLMs used for sentence simplification (in English only) to specific simplification policies. They rely on LLM-as-a-judge to avoid the need for policy-specific reference corpora. They show that their approach applied on mid-size LMs outperforms large LMs used in few-shot settings.

**Strengths:**

- convincing motivation for the work, showing the need for an approach that goes beyond dedicated seq-to-seq models as well as beyond large-scale LLMs, and that allows for adapting to a specific simplification policy
- state-of-the-art preference optimisation approach combining ARPO and SimPO
- strong baselines/toplines
- good (and somehow reassuring) results

**Weaknesses:**

- no real weakness for me, this is a good paper
- one detail: the fact that SARI has been shown to only weakly correlate with human judgement could be mentioned, and the use of SARI nevertheless could be motivated—this is one of the reasons why human evaluation is important, which the authors include in their paper
- another detail: the link between use cases and possible policies could be discussed a bit more (e.g. what about simplification targeted towards people with cognitive disabilities? when is sentence splitting necessary and when is it less useful? and so on)

**Questions:**

- am I right when I say that you use LLMs as baselines/toplines in a few-shot setting (using the prompt given in Figure 5 in the appendix)? Could you make this clearer in the main part of the paper?

---

> ### Author Response · Authors · 2025-11-21
>
> We thank the reviewer for recognizing the contribution of our work. We will try our best to incorporate the details the reviewer suggested in the revision.
>
> Regarding the question, as noted in Appendix A.2.2, we used identical zero-shot prompts for all models, because we observed that few-shot prompting sometimes led to a performance drop compared to zero-shot, for both vanilla and fine-tuned models. For example, the LENS score of the vanilla Qwen2.5-14B model on the ASSET dataset decreased by 1.5 points, and the SARI score of the Qwen2.5-7B model fine-tuned on parallel corpora decreased by 0.3 points. We conjecture that this is due to the bias introduced by few-shot examples; we therefore adopted the zero-shot setting. We will make this clear in the paper.
>
> We would like to highlight that additional experiments requested by other reviewers further confirm the effectiveness of the proposed method:
> - Our method outperforms the recent GPT-5 model (Reviewer hfEB, comment 2)
> - It applies to document-level simplification (Reviewer DkvM, comment 1)
> - It outperforms simplification with iterative feedback (Reviewer DkvM, comment 4)
> - The trained models preserve good transferability to other domains (Reviewer DkvM, comment 5)

---

### Author Response · Authors · 2025-11-26

We thank all reviewers for their feedback. We have revised the manuscript accordingly, with all newly added content highlighted in blue.
Please let us know if we address your concerns.

A summary of the key updates is provided below:

- Motivation clarification: Clarified the motivation behind our policy design in the first paragraph of the introduction.
- Further implementation details: Included computation time, prompts, and GPT-5 performance in Section 4.3. The inference settings previously located in Appendix A.1 have been moved into the main text.
- Out-of-domain sentences: see section 4.6
- Document-level simplification: see Section 5 and Appendix A.5 for results and discussion
- Comparisons with different prompt-engineering strategies: see Appendix A.4

---

### Author Response · Authors · 2025-12-02

Dear AC,

Thank you for your significant time and effort in handling our paper.

Below we summarize the reviewers’ concerns and suggestions, along with the corresponding clarifications and revisions we have made. We also indicate where each point was addressed in the rebuttal and the revised paper.

**1. Baseline selection and expansion (Reviewers:  hfEB, RRWX,  and DkvM)**

> Use of GPT-5 instead of GPT-4o (hfEB)

We conducted additional experiments and confirmed that our method outperforms the recent GPT-5 model.

_Rebuttal: hfEB Comment 2; Paper: Sec. 4.3_

> Appropriateness of baseline methods (RRWX)

The reviewer questioned our baseline selection; however, unfortunately, did not mention specific papers. We clarified that previous studies indicate that ChatGPT-based models (that we have compared with) as the state-of-the-art in sentence simplification.

_Rebuttal: RRWX Comment 4_

> More comparisons to prompt-engineering baselines (DkvM)

The reviewer suggested comparing against stronger prompting strategies such as iterative feedback. We included new experiments, and the results confirm that our method consistently outperforms this more complex prompting baseline.

_Rebuttal: DkvM Comment 4; Paper: Appendix A.4_

**2. “Preference optimization already exists” (Reviewers: hfEB, RRWX)**

Both reviewers noted that preference optimization is not new and questioned the novelty of our contribution. We clarified that our contribution is introducing a complete framework for generating policy-aligned simplifications without requiring a parallel corpus, which is a novel and unexplored technique. Preference optimization is one component within our framework, but it is not where our contribution lies.

_Rebuttals: hfEB Comment 1; RRWX Comment 2_

**3. Out-of-domain generalization (Reviewer: DkvM)**

The reviewer asked whether our method generalizes beyond Wikipedia, as both original test sets come from that domain.
We tested our trained models on datasets of news and public administration. Results show that our method transfers effectively across domains.

_Rebuttal: DkvM Comment 5; Paper: Sec. 4.6_

**4. Motivation and target users (Reviewer: DkvM)**
- Round 1 misunderstanding: The reviewer interpreted our system as general-purpose simplification and recommended user-focused design. We clarified that our method is explicitly designed for second-language learners, and the two proposed edit policies are grounded in both NLP and language-education literature.

- Round 2 concern: The reviewer suggested that sentence-level simplification for learners with special needs is a “niche” problem. We respectfully disagree. Language learning remains a major application area for text simplification, and learner needs vary significantly by proficiency level, making targeted sentence-level simplification an active and meaningful research direction.

_Rebuttal: DkvM Comment 2 and follow-up; Paper: Introduction_

**5. Document-level vs. sentence-level simplification (Reviewer: DkvM)**

The reviewer further suggested focusing on document-level rather than sentence-level simplification.

We would remark that the document-level simplification has not yet been an established problem, with a lack of appropriate datasets and evaluation metrics. Nonetheless, it is a crucial direction for future studies; we conducted preliminary experiments. The results show that our framework is promising for document-level extension.

We also clarified why sentence-level simplification is crucial for our setting:

- Pedagogical relevance: Sentence-level simplification is desirable in language learning contexts because teachers can easily track where changes have been made.
- Dependency of document-level quality: Document-level simplification requires robust sentence-level simplification, which current LLMs still struggle with. Establishing a reliable sentence-level foundation is both necessary and beneficial for future document-level work and downstream applications.

_Rebuttal: DkvM Comment 1 and follow-up; Paper: Section 5 and Appendix A.5_

---

### Meta-Review · Area_Chair_iVT4 · 2026-01-05

**Summary:**

This paper proposes a method for sentence simplification that uses "LLM-as-a-Judge" to automatically construct training data aligned with specific simplification policies (e.g., lexical-only simplification vs. full rewriting), eliminating the need for costly human-annotated parallel corpora. The approach enables small open-source LLM to match or exceed GPT-4o on simplification tasks, particularly for second-language learners at different proficiency levels.

The reviewers  had various complaints concerning methodology, the core contribution (whether it is enough), and whether ICLR is an appropriate venue for the paper. The authors did their best to address the reviewers comments, ran additional experiments, compared with additional baselines, ran out of domain experiments, even a pilot with document-level simplification.

In my view, the technical contribution of the paper is thin. I have no issue with the task itself, or the fit with ICLR. Sentence simplification is a valid task and it is interesting to see it is not yet solved by GPT. for me to be able to say that the proposed approach works, I would have to see evidence beyond SARI scores. I appreciate the authors perform a small-scale human-based evaluation but these are judged by a *single* annotator and the sample is too small to draw any meaningful conclusions (I am not sure the reported means are significantly different). I would also have liked to see sig testing or error bars for all the reported results. I also think it is not enough to create the pref alignment dataset, you should either show the method works across different languages or can be used for document-level simplification as a reviewer suggests.

**Reviewer Concerns:**

1. Limited Novelty was flagged by two reviewers who complained that the  pipeline of constructing preference pairs and applying preference learning is not new.

   The authors clarified their contribution is the complete framework for generating policy-aligned simplifications without parallel corpora, preference optimization is just one component, not the core novelty.

2. A reviewer argued that real-world needs require document-level simplification

    The authors ran preliminary document-level experiments showing their framework extends naturally. They also argued sentence-level is pedagogically valuable (teachers can track changes) and foundational for document-level work.

3. Outdated baseline (GPT-4o)

 The authors added GPT-5 experiments showing nearly identical performance to GPT-4o, and their method still outperforms GPT-5.

4. The system seems generic; should be user-focused (e.g., for people with reading impairments).

The authors clarified their method is explicitly designed for L2 learners, with policies grounded in language education literature. Reviewer DkvM maintained that this is still a "niche" problem; authors respectfully disagreed.

5. Out-of-Domain Generalization

The authors ran experiments on news (Newsela) and public administration (SimPA) domains without retraining, showing effective transfer.

6. Missing prompt engineering comparisons

The authors added experiments showing their method outperforms iterative self-correction approaches.

**Reviewer Scores:**

The scores were 8, 4, and 4. I doubt the 8 would move, the one 4 the reviewer said they retained their score. I doubt the other 4 would move upwards, but this is just a guess.

---

### Decision · Program_Chairs · 2026-01-26

Reject